

# Upper Norian conodonts from the Baoshan block, western Yunnan, southwestern China, and implications for conodont turnover

Weiping Zeng[1,2], Haishui Jiang[2], Yan Chen[2], James Ogg[2,3], Muhui Zhang[2] and Hanxinshuo Dong[2]

[1] School of Geography and Tourism, Huanggang Normal University, Huanggang, Hubei, P. R. China
[2] State Key Laboratory of Biogeology and Environmental Geology, School of Earth Sciences, China University of Geosciences, Wuhan, Hubei, China
[3] Department of Earth, Atmospheric and Planetary Sciences, Purdue University, West Lafayette, United States

Corresponding author
Haishui Jiang, jiangliuis@163.com

## ABSTRACT

The Sevatian of the late Norian is one of the key intervals in biotic turnover and in changes of paleoclimate and paleoenvironments. Conodont faunas recovered from two sections of upper Norian strata of the Dashuitang and Nanshuba formations near Baoshan City in western Yunnan province provide new insights into the diversity and biostratigraphy of the Sevatian conodonts within China as well as globally. A lower *Mockina* (*M.*) *bidentata* Zone and an upper *Parvigondolella* (*P.*) *andrusovi* Zone are identified in this area according to the first occurrences of *M. bidentata* and of *P. andrusovi*. Rich conodont fauna of *M. zapfei* is detailed and presents various intraspecific forms. A total of 19 forms of $P_1$ elements are presented, which, when combined with the reported conodonts in the *M. bidentata* Zone, suggest that there was a peak in conodont diversity within the *M. bidentata* Zone. A biotic crisis in the uppermost *M. bidentata* Zone is recognized from the contrast between the diverse conodont fauna in the *M. bidentata* Zone and the rare conodonts in the *P. andrusovi* Zone. The conodont turnover during the middle Sevatian highlights the fact that the prolonged phases of the end-Triassic mass extinction probably began in the transition interval from *M. bidentata* Zone to *P. andrusovi* Zone.

## INTRODUCTION

There is growing evidence that the end-Triassic mass extinction episode was a prolonged interval comprised of multiple waves of extinctions (*Benton, 1986*, *1993*; *Hallam, 2002*; *Bambach, Knoll & Wang, 2004*; *Tanner, Lucas & Chapman, 2004*; *Ward et al., 2004*; *Lucas & Tanner, 2008*; *Onoue et al., 2016*; *Onoue, Hori & Kojima, 2017*; *Rigo et al., 2020*; *Wignall & Atkinson, 2020*; *Racki & Lucas, 2020*; *Lucas, 2021*). This protracted Late Triassic extinction episode probably begin as early as the middle or late Norian (*Sephton et al.,*

*2002; Tanner, Lucas & Chapman, 2004; Lucas & Tanner, 2015; Rigo et al., 2020*). For example, the global diversity of marine and terrestrial tetrapods peaked during the late Carnian, then dropped steadily from early Norian through late Rhaetian to reach a nadir at the end of the Triassic (*Benson et al., 2009; Benton et al., 2013*). Almost all monotid bivalves disappeared around the Norian-Rhaetian boundary (*Wignall et al., 2007*), and significant turnovers in ammonoid and conodont faunas occurred during the late Norian (*McRoberts, Krystyn & Shea, 2008*) and during the Norian-Rhaetian boundary interval (*Ward et al., 2001*). According to the compilation of *Benton (1993)*, seven families of gastropods had gone extinct by the end of the Norian, whereas none underwent an extinction during the end-Triassic and 35 families continued into the Jurassic. *Onoue et al. (2016)* concluded that a succession of radiolarian extinctions took place at the mid-Norian, the Norian-Rhaetian boundary interval and the Triassic-Jurassic boundary. *Rigo et al. (2020)* placed the onset of the prolonged set of Late Triassic mass extinctions as at or very close to the Norian/Rhaetian boundary based on $\delta^{13}C_{org}$ data from sections distributed around the world. Other evidence suggests that the onset was slightly earlier during the Sevatian.

During the Sevatian of the Late Norian, many genera underwent significant turnover events or extinctions and there were significant changes in paleoclimate and paleoenvironments. *McRoberts (2010)* observed that almost all of the bivalves of the *Halobia* and *Eomonotis* genera disappeared and those of *Monotis* began to appear during the Alaunian/Sevatian transition (ca. 217–213 Ma). *Baranyi et al. (2018)* quantitatively analyzed the pollen data of the Chinle Formation in the southwestern United States and found that turnovers in plants also occurred during this Alaunian/Sevatian transition, in addition to other significant changes in flora and fauna in the middle of the Chinle Formation. Qualitative and quantitative analysis of sporopollen in Poland documented a shift to hygrophytes during the Sevatian, indicating a small change to more humid climates (*Mader, 2015*). *Benton (1986)* proposed that diversity of genera of ammonoids declined sharply during the Late Norian. *Wiedmann & Kullmann (1996)* further pointed out about 14 genera disappeared at the Alaunian/Sevatian boundary followed by a further progressive or, at least gradual, decline in diversity of ammonoids during the Sevatian through Rhaetian. *Lucas (2018)* summarized previous studies on ammonoids and suggested that, as the diversity of ammonoids decreased, heteromorphs appeared during the Sevatian. *Onoue et al. (2016)*, *Onoue, Hori & Kojima (2017)* observed a gradual extinction of radiolarians in the "*Epigondolella*" (=*Mockina*) *bidentata* Zone of the Sevatian substage in the Sakahogi section of Panthalassa, and a short negative shift of $^{187}Os/^{188}Os$ and $\delta^{13}C_{org}$ values in the lower *Mockina bidentata* Zone. In contrast, calcareous ultra-microplankton expanded and diversified in the *Mockina bidentata* Zone of the Sevatian (*Gardin et al., 2012; Preto et al., 2013; Demangel et al., 2020*).

In terms of the climatic and environmental trends, the curve of $^{18}O_{phosphate}$ from Triassic conodonts shows a prominent negative shift in the lower Sevatian, which indicates an interval ("W3") of relatively warm and humid climate (*Trotter et al., 2015*), even though proxies from paleosol carbonate rocks in the Newark and Hartford basins of the United States suggest a significant decrease in atmospheric $pCO_2$ concentration during the Sevatian (*Schaller, Wright & Kent, 2015; Kent, Olsen & Muttoni, 2017*). *Zaffani et al. (2017)*

identified three negatively biased events (S1, S2 and S3) by the Sevatian portion of the $\delta^{13}C_{org}$ curves of the Pignola-Abriola, the Mt Volturino and the Madonna del Sirino sections in southern Italy, which enhanced the broader negative shift in $\delta^{13}C_{carb}$ recorded by *Muttoni et al. (2014)*. There was a rapid northward dispersal of sauropods from Gondwana to temperate Europe and Greenland (215–212 Ma, *Kent & Clemmensen, 2021*). The Manicouagan impact event (215–214 Ma, *van Soest et al., 2011*; *Sato et al., 2021*) in Canada might have punctuated these climatic trends.

Therefore, the Sevatian substage of the Norian is one of the key intervals in biological evolution within the Late Triassic and heralds the onset of the series of end-Triassic mass extinctions. The major changes in biology and climate occurred during the conodont *M. bidentata* Zone or in the transition interval of *M. bidentata* Zone/*P. andrusovi* zone. Conodonts, prior to their total extinction in the latest Triassic, have advantages in the study of the biotic events and reconstruction of precise biostratigraphy during the Triassic because of their rapid evolution, wide distribution and well-preserved characteristics. Nevertheless, conodonts of the Sevatian have been poorly studied; indeed, current research records suggest an anomalously low rate of conodont evolution during the Sevatian (*Orchard, 2018*; *Rigo et al., 2018*). In particular, it appears that the duration of the single *M. bidentata* Zone was more than 5 Myr, which is inconsistent with the typical fast-evolving nature of these conodont animals (*e.g. Mosher, 1968*; *Bergström, 1983*; *Miller & Clark, 1984*; *Orchard, 2007*, *2018*; *Chen et al., 2019*). However, evolving and disputed concepts of the classification and identification of conodonts of Sevatian age (*e.g. Orchard, 1991b*, *2018*; *Dong & Wang, 2006*; *Rigo et al., 2018*; *Karádi et al., 2021*) might have distorted this picture.

## Development and complexity of Sevatian conodont biostratigraphy

*Huckriede (1958)* established a vaguely defined form species of "*Polygnathus*" *abneptis*, which contains various denticulated $P_1$ elements, and some of these denticulated specimens occurred in the Sevatian substage of the Mediterranean. *Mosher (1968)* established *M. bidentata* and summarized that "*Epigondolella* (*E.*)" *abneptis* and *Norigondolella* (*N.*) *steinbergensis* co-occurred with *M. bidentata* in Europe, whereas only "*E.*" *abneptis* occurred in the *M. bidentata* Zone in North America. However, because the original definition of "*E.*" *abneptis* was based on a variety of platform conodonts (*Huckriede, 1958*; *Karádi, 2018*), "*E.*" *abneptis* actually includes many different forms of $P_1$ elements, which indicates a more diverse suite of conodont fauna in *M. bidentata* Zone. *Kozur & Mostler (1972*, table 1) suspected that *M. postera* may range into the Sevatian of the Tethyan Triassic strata of Europe (excluding the Far Mediterranean Basin and Greece). *Kovács & Kozur (1980*, table 2) presented stratigraphic ranges for the most important Middle and Upper Triassic conodonts, and indicated that *M. mosheri, N. steinbergensis, M. postera, M. longidentata* and *O? multidentata* occurred in *M. bidentata* Zone. *Krystyn (1980*, figs. 6, 8) also compiled that *M. postera*, "*E.*" *abneptis* and *N. steinbergensis* range into *M. bidentata* Zone. The taxa of *M. postera* (*Wang & Dong, 1985*; *Gullo, 1996*; *Channell et al., 2003*; *Muttoni et al., 2004*; *Rožič, Kolar-Jurkovšek & Šmuc, 2009*) and *N. steinbergensis* (*Channell et al., 2003*; *Hornung, 2005*; *Rožič, Kolar-Jurkovšek & Šmuc,*

*2009*; *Mazza, Rigo & Gullo, 2012*; *Onoue et al., 2018*; *Du et al., 2020*) are reported in many places in the Tethyan realm. *Wang & Dong (1985)* first presented eastern Tethyan platform conodonts of Norian age in Baoshan area, Yunnan Province of China, and showed that *M. postera*, "E." *abneptis spatulatus*, "E" *multidentata* and "E." *abneptis abneptis* are associated with *M. bidentata*.

Many other form species accompanying *M. bidentata* in the Tethys realm have been discovered during the 21$^{st}$ century. *M. zapfei* (*Channell et al., 2003*; *Giordano et al., 2010*; *Rigo et al., 2018*; *Du et al., 2021*; *Jin et al., 2022*), *M. slovakensis* (*Gullo, 1996*; *Giordano et al., 2010*; *Muttoni et al., 2004*; *Rigo et al., 2018*; *Du et al., 2021*; *Jin et al., 2022*) and "P." *vrielyncki* (*Channell et al., 2003*; *Rigo et al., 2018*; *Du et al., 2021*) are common species that occur in the *M. bidentata* Zone. "M." *englandi* (*Krystyn et al., 2007*; *Onoue et al., 2018*; *Du et al., 2020*) and "M." *carinata* (*Du et al., 2020*) are also reported. *Karádi (2021)* displayed "Orchardella" *mosheri* morphotype B and *M. englandi* from Hungary, but did not state whether the P$_1$ elements of the two form species were found in the *M. bidentata* Zone. Some different forms of P$_1$ elements identified as *M. mosheri* morphotype B or morphotype A (*Du et al., 2020*, fig. 3.8; *Krystyn et al., 2007*, pl. 1, fig. 6; *Jin et al., 2022*, fig. 4.8) occur in the *M. bidentata* Zone. Segminate P$_1$ elements of *P. lata* (*Du et al., 2021*), conical P$_1$ elements of *Zieglericonus* (*Channell et al., 2003*; *Du et al., 2021*) and transitional forms with just one marginal denticle and no or extremely reduced platform are found in the *M. bidentata* Zone (*Karádi et al., 2020*; *Du et al., 2021*; *Zeng et al., 2021*). In addition, many undefined species or forms of P$_1$ elements are documented in the *M. bidentata* Zone, such as *M.* cf. *zapfei* (*Channell et al., 2003*), *M. carinata*? (*Du et al., 2020*, fig. 3.9), *M.* cf. *slovakensis* (*Channell et al., 2003*), *E. triangularis*? (*Channell et al., 2003*), *Mockina* sp. (*Du et al., 2020*), *M.* aff. *tozeri* (*Mazza, Rigo & Gullo, 2012*; *Onoue et al., 2018*), *E. uniformis*? (*Mazza, Rigo & Gullo, 2012*; *Onoue et al., 2018*), *etc. Zeng et al.* (2021, fig. 2.3) first found *M. sakurae* and *Jin et al. (2022)* erected a new species *M. passerii* Rigo & Du, 2022 in the *M. bidentata* Zone of the Baoshan area, China.

In North America, there are only a few reported association form species in the *M. bidentata* Zone; mainly these are *E. englandi* (*Krystyn et al., 2007*; *Onoue et al., 2018*; *Du et al., 2020*), *E. carinata* (*Du et al., 2020*), *N. steinbergensis* and unidentified species of *Parvigondolella* (*Orchard, 1991b*; *Orchard et al., 2007b*). In Japan of the Panthalassa realm, *Yamashita et al. (2018)* figured eight different forms of P$_1$ elements from the *M. bidentata* Zone, and respectively assigned these to *M. spiculata*, *M. elongata*, *M. mosheri* A, *M. slovakensis*, *Mockina* sp. indet. A, *Mockina* sp. indet. B and *P.* aff. *vrielyncki*.

The overlying *P. andrusovi* Zone was first introduced by *Kovács & Kozur (1980)*, who put it above the *M. bidentata* Zone and below the *Mi. hernsteini* Zone and with an interpreted age corresponding to the middle of the ammonoid *Cochloceras suessi* Zone. *Gaździcki, Kozur & Mock (1979)* used these three conodont biozones to subdivide Sevatian strata of the Alpine-Mediterranean Triassic. Later, *Kozur (2003)*, *Channell et al. (2003)*, *Rožič, Kolar-Jurkovšek & Šmuc (2009)* and *Gale et al. (2012)* combined the *P. andrusovi* and *Mi. hernsteini* into one single biozone. At present, *P. andrusovi* Zone is widely recognized in the Tethyan realm (*Rigo et al., 2018*; *Karádi et al., 2020*; *Du et al., 2021*; *Zeng et al., 2021*). *Yamashita et al. (2018)* discovered *P. andrusovi* in Japan. In North

America, despite no *P. andrusovi* having been reported, several species of *Parvigondolella* occur from the upper Norian to the Rhaetian (*Carter & Orchard, 2007*; *Orchard et al., 2007a*, *2007b*).

This overview indicates that conodont form species are more abundant and varied in the *M. bidentata* Zone of the early Sevatian than initially thought, and that more detailed research of the fauna is required in order to enhance the database for conodont biostratigraphy, evolution and diversity. It is essential that such studies include a more precise intercalibration to other biostratigraphic, geochemical and magnetostratigraphic scales. China, located in the eastern Tethys, has Sevatian strata distributed in Heilongjiang of northeastern China (Table 1, *Wang, Kang & Zhang, 1986*; *Wang & Wang, 2016*), Tibet of southwestern China (Table 1, *Mao & Tian, 1987*; *Yi et al., 2003*; *Ji et al., 2003*) and Western Yunnan of southwestern China (Table 1, *Wang & Dong, 1985*; *Dong & Wang, 2006*; *Du et al., 2020*; *Zeng et al., 2021*; *Jin et al., 2022*). However, the investigation on Norian conodonts in these locations is very limited and many conodont taxa are not revised (Table 1). For example, the presented specimens identified as "*E.*" *multidentata* in the documentations given in Table 1 actually all have different morphological features from *O? multidentata*. For example, the specimen (*Wang & Dong, 1985*, pl. 1, fig. 16) identified as "*E.*" *multidentata* has three and one marginal denticles, respectively, on each anterior platform margin, which more resembles *M. zapfei* (see entry below in Systematic paleontology), and the other specimen of "*E.*" *multidentata* (*Wang & Dong, 1985*, pl. 1, fig. 17) differs from *O? multidentata* by having only one inner anterior marginal denticle and denticulated posterior platform margins. As our article mainly focuses on the conodont diversity during the Norian, especially the Sevatian, we don't discuss the accuracy of conodont classification and hence retain the authors' original taxonomy in Table 1.

Only a few studies have focused on the late Norian conodonts of the Baoshan area. The first by *Wang & Dong (1985)* established an "*E.*" *postera* Zone and "*E.*" *bidentata* Zone, and assigned most of the Dashuitang Formation to their "*E.*" *postera* Zone and the upper Dashuitang Formation and the lower Nanshuba Formation to their "*E.*" *bidentata* Zone. Even though the P$_1$ elements illustrated in *Wang & Dong (1985)* only present a single view, at least eight different forms in upper view can be discerned. Late, *Wang et al. (2019)* discovered "*M.*" *englandi* (*Orchard, 1991b*), "*M.*" aff. *englandi* and *M. bidentata* in the Dashuitang Formation. *Du et al. (2020)* presented several additional forms of Norian conodonts from the Nanshuba Formation. *Zeng et al. (2021)* illustrated *M. sakurae* (fig. 2.3), *P. andrusovi* (figs. 1/e, 4/3), P$_1$ elements with only one marginal denticle and no platform (figs. 5.2h, 5.7) and P$_1$ elements with only a pair of anterior marginal denticles and squared, smooth posterior platform (figs. 4.2b–d) for the first time from the Baoshan area. *Jin et al. (2022)* first reported *M. slovakensis* from the Baoshan area and erected *M. passerii* Rigo & Du, 2022. These studies imply that the conodonts in the Upper Triassic strata of the Baoshan block are probably very diverse.

Therefore, because the Baoshan area of western Yunnan has yielded relatively diverse late Norian conodonts (*Wang & Dong, 1985*; *Du et al., 2020*; *Zeng et al., 2021*; *Jin et al., 2022*) and has easy access, we performed a detailed study of conodonts in two sections in

**Table 1  Reported conodont zonation of the Norian stage in China and its global correlation.**

| Substage | Wang & Wang, 1976, Tingri, Tibet | Qiu, 1984, Lhasa, Tibet | Tian, 1982, Nyalam, Tibet | Mao & Tian, 1987, Lhasa, Tibet | Wang & Wang, 1990, Yushu, Qinghai | Wang, 1993, Changdu, Tibet; Yidun, Sichuan | Yi et al., 2003, Qiangtang, Tibet | Ji et al., 2003, Lhasa, Tibet | Wang, Kang & Zhang, 1986, Nada Hadan, Heilongjiang | Wang & Dong, 1985, Baoshan, Yunnan | Jin et al., 2022, Baoshan, Yunnan | This study, Baoshan, Yunnan | Dong & Wang, 2006, Yunnan | Wang & Wang, 2016, China | Rigo et al., 2018, Tethys — Conodont Zones | Rigo et al., 2018, Tethys — Ammonoid Zones | Orchard et al., 2007a, 2018, North America — Conodont Zones | Orchard et al., 2007a, 2018, North America — Ammonoid Zones |
|---|---|---|---|---|---|---|---|---|---|---|---|---|---|---|---|---|---|---|
| Sevatian | E. abneptis | E. abneptis | E. multi. | E. bidentata | | | E. bidentata -E. sp. | unclear | P. andrusovi | E. bidentata | M. bidentata | P. andrusovi / M. bidentata | Mi. posthern. / Mi. hernsteini -P. andrusovi / E. bidentata | Mi. hernsteini / P. andrusovi / E. bidentata | Mi. posthern. / Mi. hernsteini / P. andrusovi / M. bidentata | S. quiquepunctatus | mosheri / bidentata | Gn. cordilleranus / Mes. columbianus |
| Alaunian | | | | E. postera | E. postera-abneptis abneptis spatulatus | E. postera- E. abneptis | E. postera -abneptis spatulatus | E. bidentata / unclear / E. postera | E. bidentata / E. postera | E. / postera | M. slovakensis | | E. postera | E. postera | M. slovakensis / M. serrulata / M. postera / M. spiculata | Ha. macer / Hi. hogarti | serrulata / postera / elongata | |
| Lacian | E. abneptis | E. abneptis | E. abneptis | E. sp. C / E. multi. | | E. abneptis | | E. tozeri / E. spiculata / unclear / ?E. tri. / unclear / ?E. primitia | E. multi. / E. abneptis | | | | E. multi. / E. abneptis / E. pseudodiebeli -E. abneptis | E. multi. / E. abneptis | M. spiculata / E. rigo-Equadrata / C. gulloae / Me. parvus | Cy. bicrenatus / J. magnus / Ma. paulckei / G. jandianus | spiculata / tozeri / multi. / transformis / triangularis / quadrata / primitia | D. rutherfordi / ? / J. magnus / Ma. dawsoni / St. kerri |

**Note:**

E., Epigondolella; M., Mockina; P., Parvigondolella; Mi., Misikella; Me., Metapolygnathus; C., Carnepigondolella; tri., triangularis; multi., multidentata; posthern., posthernsteini; G., Guembelites; Ma., Malayites; J., Juvavites; Cy., Cyrtopleurites; Hi., Himavatites; Ha., Halorites; S., Sagenites; St., Stikinoceras; D., Drepanites; Mes., Mesohimavatites; Gn., Gnomohalorites.

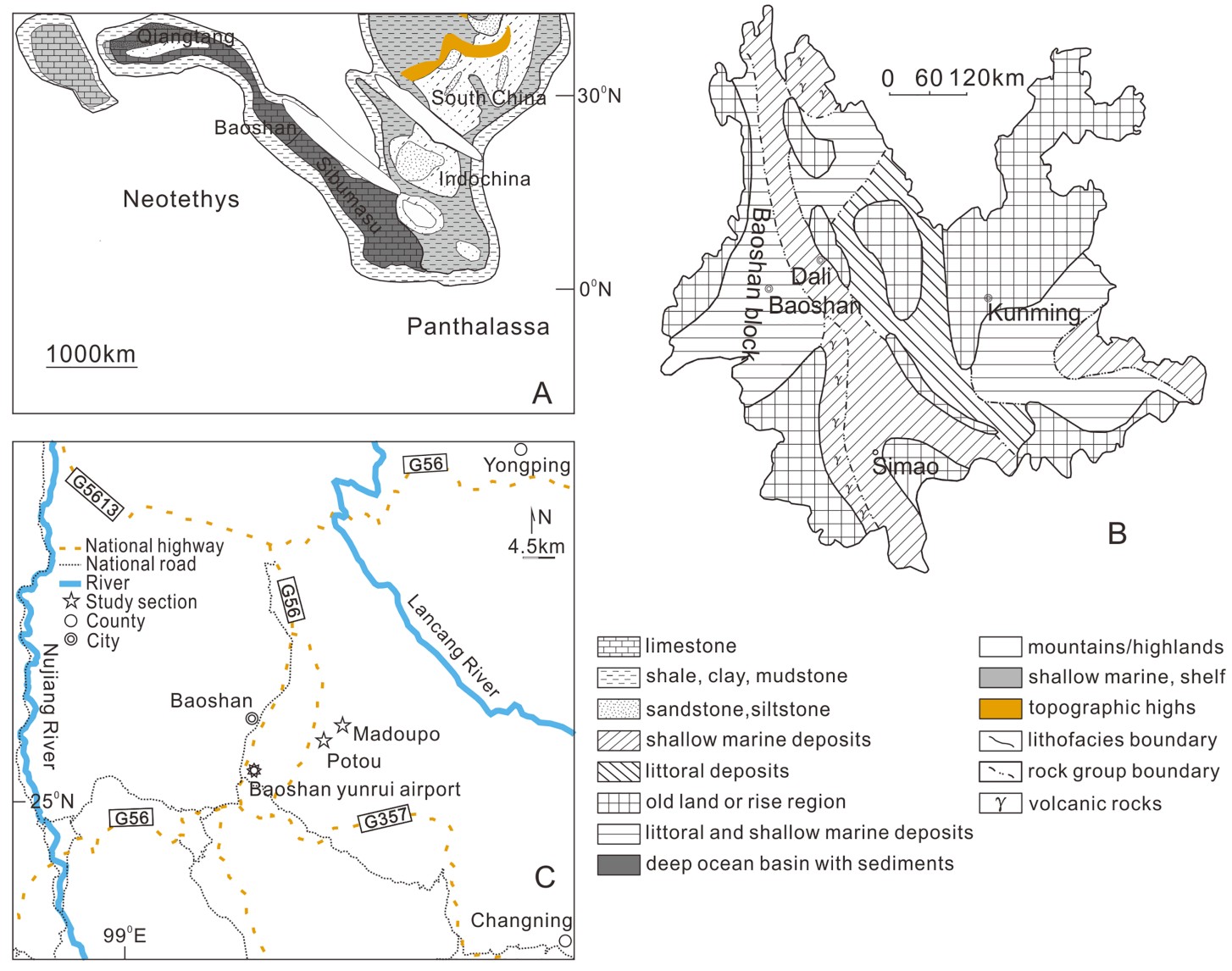

**Figure 1 Geographic locations of the studied sections.** (A) Paleogeographic map (after *Golonka, Embry & Krobicki, 2018*) showing the position of the Baoshan block during Late Triassic prior to accretion to South China. (B) Late Triassic paleogeography map of Yunnan in modern coordinates (modified from *Bureau of Geology and Mineral Resources of Yunnan Province, 1990*) showing the position and the depositional facies of the Baoshan block. (C) Map showing the locations of the described sections.

Baoshan to acquire a detailed set of data to gain a better understanding of the Sevatian conodonts of eastern Tethys.

## Geological setting

The conodonts described in this article were recovered from the Madoupo section (MDP) and the Potou section (POT), which are located about 5 to 6 km apart in the Longyang district, Baoshan city, western Yunnan Province, southwestern China (Fig. 1C). These two sections are situated in the Baoshan tectonic block, which is the current northern part of the Sibumasu terrane (Fig. 1A, *Ali et al., 2013*; *Liao et al., 2015*; *Cai et al., 2017*) and may

have accreted to South China during the Late Triassic or Early Jurassic (*Sone & Metcalfe, 2008*; *Morley, 2018*). The upper Triassic marine sediments of the Baoshan block are well developed (Figs. 1A, 1B), and can be subdivided into the Dashuitang Formation and the Nanshuba Formation (*Li, 1976*; *Bureau of Geology and Mineral Resources of Yunnan Province, 1980*, *1981*; *Wang & Dong, 1985*; *Zhao et al., 2012*; *Wang et al., 2019*; *Du et al., 2020*; *Jin et al., 2022*). The ages of the outcropping limestones of the Dashuitang Formation and of the Nanshuba Formation are constrained by conodonts and radiolarians as Alaunian-Sevatian and as Sevatian respectively (*Wang & Dong, 1985*; *Wang et al., 2019*; *Du et al., 2020*; *Jin et al., 2022*).

The interpretations of the depositional environments of the Dashuitang and Nanshuba formations are inconclusive. The earliest studies inferred that the Upper Triassic strata on the Baoshan block were deposited in littoral to shallow-marine environments (Fig. 1B) (*Bureau of Geology and Mineral Resources of Yunnan Province, 1980*, *1990*). *Hao (1999)* thought that the limestone of the Dashuitang Formation was a shallow carbonate platform facies. *Bao et al. (2012)* discovered seismites in the Dashuitang Formation in the Jinji area of Baoshan, and therefore concluded that it deposited in a slope-basin environment. *Peng, Huang & Yuang (2014)* found carbonate-clastic turbidites in the Dashuitang Formation at Yaoguan of western Yunnan, and inferred that it was deposited in a rifted trough basin. *Wang et al. (2019)* deduced from the microfacies of the upper limestone of the Dashuitang Formation in the Dabaozi area of Baoshan that the setting was a continental shelf environment. *Wu et al. (2020)* subdivided the Dashuitang Formation deposits in the Hongyan section into eight microfacies, and distinguished three different depositional settings—deep-water shelf facies, slope facies, and base-of-slope facies—within a generally relatively low-energy deep-water environment, which corresponded to an extensive transgression during Late Triassic across northwestern Yunnan.

For the overlying Nanshuba Formation, most researchers agree that it was deposited in deeper waters than the Dashuitang Formation, but dispute whether the Nanshuba Formation was deposited on the marginal part of a slope-basin (*Zhao et al., 2012*), on the marginal part of a rifting seaway (*Wang, Li & Duan, 2000*), or on the shelf of a shallow island arc with steep slopes (*Wang et al., 2019*). The Nanshuba Formation in Baoshan area is dominated by calcareous mudstone and sandy mudstone but with a few interbeds of marl and limestone. Regionally, the total thickness of this formation ranges between 800–1,400 m, but varies dramatically laterally. The thickness of the interbedded limestones can be up to 100 m (*Bureau of Geology and Mineral Resources of Yunnan Province, 1990*, p. 194).

The lower ca. 13 m of the Potou section is mainly composed of grayish yellow thick-bedded bioclastic limestone and greyish white medium-bedded limestone of the upper Dashuitang Formation. The exposed strata are slightly fragmented and weathered on the surface of the section. The Dashuitang Formation at the Potou section conformably underlies the Nanshuba Formation, which was not well exposed due to two significant covered intervals and with herbage and low shrubs hiding other parts. The lower part of the Nanshuba Formation exposed below the first covered interval consists of thin- to medium-bedded grey to white limestone and an upper greyish green shale interbedded with grayish-yellow marl. The upper parts of the Nanshuba Formation exposed at the

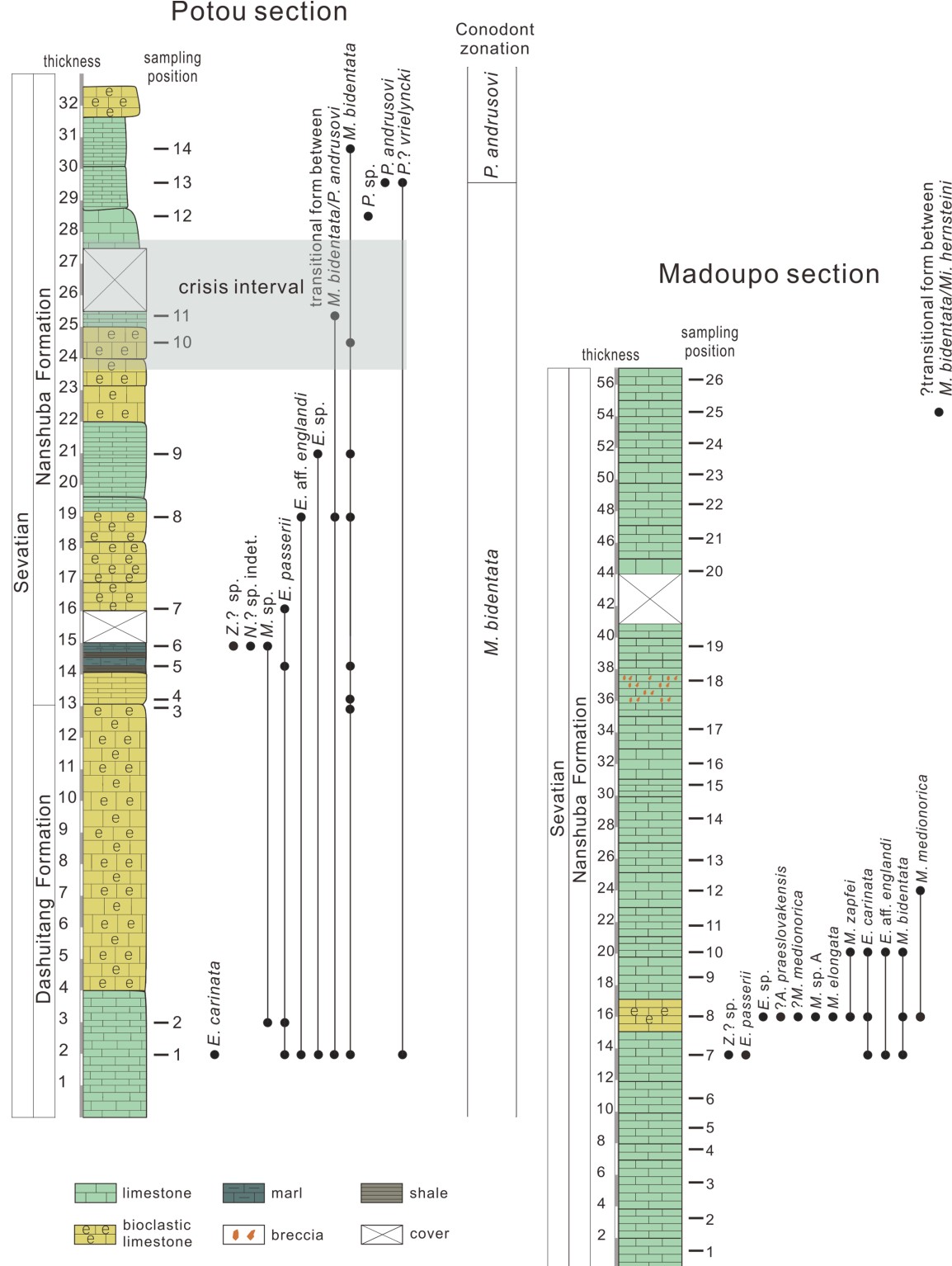

**Figure 2 Conodont distribution in the Potou section and in the Madoupo section, Baoshan city, Yunnan Province, southwestern China.**

**Table 2 Stratigraphic distribution and statistic results of conodont species from the Potou section and the Madoupo section.**

| Sample no. | ?A. praeslovakensis | E. carinata | E. aff. englandi | E. passerii | E. sp. | M. bidentata | M. elongata | M. medionorica | ?M. medionorica | M. zapfei | Mockina sp. A | transitional form between M. bidentata/P. andrusovi | ?transitional form between M. bidentata/Mi. hernsteini | M. sp. | P. andrusovi | P. sp. | P.? vrielyncki | N.? sp. indet. | Z.? sp. | Unidentifiable P$_1$ elements | S elements |
|---|---|---|---|---|---|---|---|---|---|---|---|---|---|---|---|---|---|---|---|---|---|
| POT14 | | | | | | 1 | | | | | | | | | | | | | | 1 | |
| POT13 | | | | | | | | | | | | | | | | 1 | 1 | | | | |
| POT12 | | | | | | | | | | | | | | | | | 1 | | | 1 | 3 |
| POT11 | | | | | | | | | | | | 1 | | | | | | | | 1 | |
| POT10 | | | | | | 1 | | | | | | | | | | | | | | 1 | 4 |
| POT9 | | | | | 1 | 1 | | | | | | | | | | | | | | 4 | |
| POT8 | | | 1 | | | 2 | | | | | | 1 | | | | | | | | 3 | |
| POT7 | | | | 1 | | | | | | | | | | | | | | | | 2 | 1 |
| POT6 | | | | | | | | | | | | | | | 1 | | | 1 | 1 | 4 | |
| POT5 | | | | 1 | | 2 | | | | | | | | | | | | | | | |
| POT4 | | | | | | 1 | | | | | | | | | | | | | | | |
| POT3 | | | | | | 1 | | | | | | | | | | | | | | 1 | |
| POT2 | | | | 1 | | | | | | | | | | | 3 | | | | | | |
| POT1 | | 2 | 5 | 3 | 2 | 11 | | | | | | 3 | | | | | 1 | | | | |
| MDP26 | | | | | | | | | | | | | | | | | | | | | |
| MDP25 | | | | | | | | | | | | | 1 | | | | | | | | |
| MDP24 | | | | | | | | | | | | | | | | | | | | | |
| MDP23 | | | | | | | | | | | | | | | | | | | | | |
| MDP22 | | | | | | | | | | | | | | | | | | | | | |
| MDP21 | | | | | | | | | | | | | | | | | | | | | |
| MDP20 | | | | | | | | | | | | | | | | | | | | | |
| MDP19 | | | | | | | | | | | | | | | | | | | | | |
| MDP18 | | | | | | | | | | | | | | | | | | | | | |
| MDP17 | | | | | | | | | | | | | | | | | | | | | |
| MDP16 | | | | | | | | | | | | | | | | | | | | | |
| MDP15 | | | | | | | | | | | | | | | | | | | | | |
| MDP14 | | | | | | | | | | | | | | | | | | | | | |
| MDP13 | | | | | | | | | | | | | | | | | | | | | |
| MDP12 | | | | | | | 1 | | | | | | | | | | | | | | |
| MDP11 | | | | | | | | | | | | | | | | | | | | | |
| MDP10 | | 2 | 1 | | | 3 | | | | 1 | | | | | | | | | | 4 | |
| MDP9 | | | | | | | | | | | | | | | | | | | | 4 | |
| MDP8 | 1 | 2 | | | 1 | 2 | 28 | 1 | 1 | 59 | 3 | | | | | | | | | 77 | 25 |
| MDP7 | | 5 | 2 | 4 | | 6 | | | | | | | | | | | | | 1 | 15 | 3 |
| MDP6 | | | | | | | | | | | | | | | | | | | | | |
| MDP5 | | | | | | | | | | | | | | | | | | | | | |

| Sample no. | ?A. praeslovakensis | E. carinata | E. aff. englandi | E. passerii | E. sp. | M. bidentata | M. elongata | M. medionorica | ?M. medionorica | M. zapfei | Mockina sp. A | transitional form between M. bidentata/ P. andrusovi | ?transitional form between M. bidentata/ Mi. hernsteini | M. sp. | P. andrusovi | P. sp. | P.? vrielyncki | N.? sp. indet. | Z.? sp. | Unidentifiable P$_1$ elements | S elements |
|---|---|---|---|---|---|---|---|---|---|---|---|---|---|---|---|---|---|---|---|---|---|
| MDP4 | | | | | | | | | | | | | | | | | | | | | |
| MDP3 | | | | | | | | | | | | | | | | | | | | | |
| MDP2 | | | | | | | | | | | | | | | | | | | | 1 | |
| MDP1 | | | | | | | | | | | | | | | | | | | | | |

**Note:**

*A., Ancyrogondolella; E., Epigondolella; M., Mockina; N., Norigondolella; P., Parvigondolella; Z., Zieglericonus; Mi., Misikella; POT, Potou section; MDP, Madoupo section.*

Potou section mainly consist of medium-bedded greyish white bioclastic limestone and micritic limestone. The Madoupo section is relatively continuous, ca. 56-m exposure, with the lower part along a hiking road transitioning to the upper part at the top of the hill. It mainly consists of grey micritic limestone with a few beds of bioclastic limestone. The intervals of the Nanshuba Formation studied by *Wang et al. (2019)* and by *Du et al. (2020)* could be stratigraphically lower than our studied section.

## MATERIALS AND METHODS

Limestone samples at the Baoshan block were collected from the Madoupo section (26 samples) and the Potou section (14 samples) (for sampling positions see Fig. 2). The weight of each sample from the Potou section ranges between 5 and 10 kg. No precise weights were made of the samples from the Madoupo section, but each sample weighed no less than ~5 kg. The samples were crushed into small pieces and processed in a 10% solution of acetic acid. The process of extracting conodonts is detailed in *Jiang et al. (2019)* and *Yuan, Jiang & Wang (2015)*.

Six samples of the Madoupo section yielded conodonts (Table 2), but only five of the samples (MDP7, MDP8, MDP10, MDP12 and MDP25) yielded identifiable P$_1$ elements and these show a Color Alteration Index (CAI) of 1–1.5. A total of 126 identifiable P$_1$ elements were obtained from the Madoupo section, most of which were collected from the sample MDP8. All samples from the Potou section yielded conodonts, and these show a CAI value of 1–2. A total of 52 identifiable P$_1$ elements were collected. The collected specimens were photographed using a scanning electron microscope (SEM).

*Repository and institutional abbreviation*—All conodonts examined in this study are deposited in the School of Earth Sciences, China University of Geosciences, Wuhan, Hubei, P.R. China.

## RESULTS

The occurrence and distribution of conodont taxa in each bed are shown in Fig. 2 and Table 2. Two conodont zones are discriminated based on the first occurrences (FO) of
*M. bidentata* and of *P. andrusovi*. Marker species of *M. bidentata* are distributed nearly throughout the entire Potou section. The lower part of the Madoupo section that yielded conodonts also recovered *M. bidentata*. *P. andrusovi* first occurs at the ~29.5 m level in the Potou section above the appearance of a transitional form between *M. bidentata* and *P. andrusovi* and *Parvigondolella* sp. Taking the layer with the lowest occurrence of *P. andrusovi* as the zonal boundary, then the Potou section can be divided into a *M. bidentata* Zone and a *P. andrusovi* Zone. As only one specimen of a ?transitional form between *M. bidentata* and *Mi. hernsteini*, which is described below, was found in the uppermost part of the Madoupo section, then it is inferred that the entire Madoupo section is still within the *M. bidentata* Zone.

**Mockina bidentata Zone**—Lower limit: the first occurrence (FO) of *M. bidentata*. Upper limit: the FO of *P. andrusovi*.

Associated taxa in the Potou section: *E. carinata*, *E.* aff. *englandi*, *E. passerii*, *Epigondolella* sp., *Mockina* sp., *Norigondolella*? sp. indet., *Parvigondolella* sp., *P.*? *vrielyncki*, *Zieglericonus*? sp. and transitional form between *M. bidentata* and *P. andrusovi*. A total of 10 different forms of $P_1$ elements occurred in *M. bidentata* Zone. One broken specimen presents characterizations of genus *Norigondolella* (Figs. 3PP–3QQ), with flat and unornamented platform margins which extend to or near the anterior end and may have intense microcrenulation, with an narrow groove and with laterally compressed denticles which are fused in the lower parts and are separated near the tips. Five $P_1$ elements has only one marginal denticle and no platform, which were also presented in the *P. andrusovi* Zone of the Xiquelin section in Baoshan area (*Zeng et al., 2021*). *Karádi et al. (2020)* and *Du et al. (2021)* illustrated in detail that this form of $P_1$ elements is a transitional form between *M. bidentata* and *P. andrusovi*.

Associated taxa in the Madoupo section: ?*Ancyrogondolella* (*A.*) *praeslovakensis*, *E. carinata*, *E.* aff. *englandi*, *E. passerii*, *Epigondolella* sp., *M. elongata*, *M. medionorica*, ? *M. medionorica*, *M. zapfei*, *Mockina* sp. A, *Zieglericonus*? sp. and a probable transitional form between *M. bidentata* and *Mi. hernsteini*. A total of 12 different forms of $P_1$ elements are found.

One broken $P_1$ element has sub-symmetrically bifurcated keel end, denticulated anterior platform, smooth posterior platform margins, anterior-located cusp, submedian pit, highly fused blade denticles, abrupt blade end and peculiar arched lateral profile of the base (Figs. 3MM–3OO), which are identical with *A. praeslovakensis*. However, the classification of this $P_1$ element can't be totally confirmed due to the broken posterior platform. *M. medionorica* previously was commonly recovered from the Alaunian of the western Tethys (*Kovács & Kozur, 1980*; *Vrielynck, 1987*; *Kozur, 2003*; *Channell et al., 2003*; *Karádi et al., 2021*) and of the Panthalassa (*Ishida & Hirsch, 2001*), its occurrence in the lower Sevatian of the Madoupo section in eastern Tethys indicates that it has a longer range and wider distribution.

Both sections yield *E. carinata*, *E.* aff. *englandi*, *E. passerii* and *Epigondolella* sp. One conic element from the Madoupo section bears a small node above the anterior base (Figs. 4T–4U), the other conic element from the Potou section possesses a small node above the posterior base (Figs. 4V–4W), and both conic elements have a normally

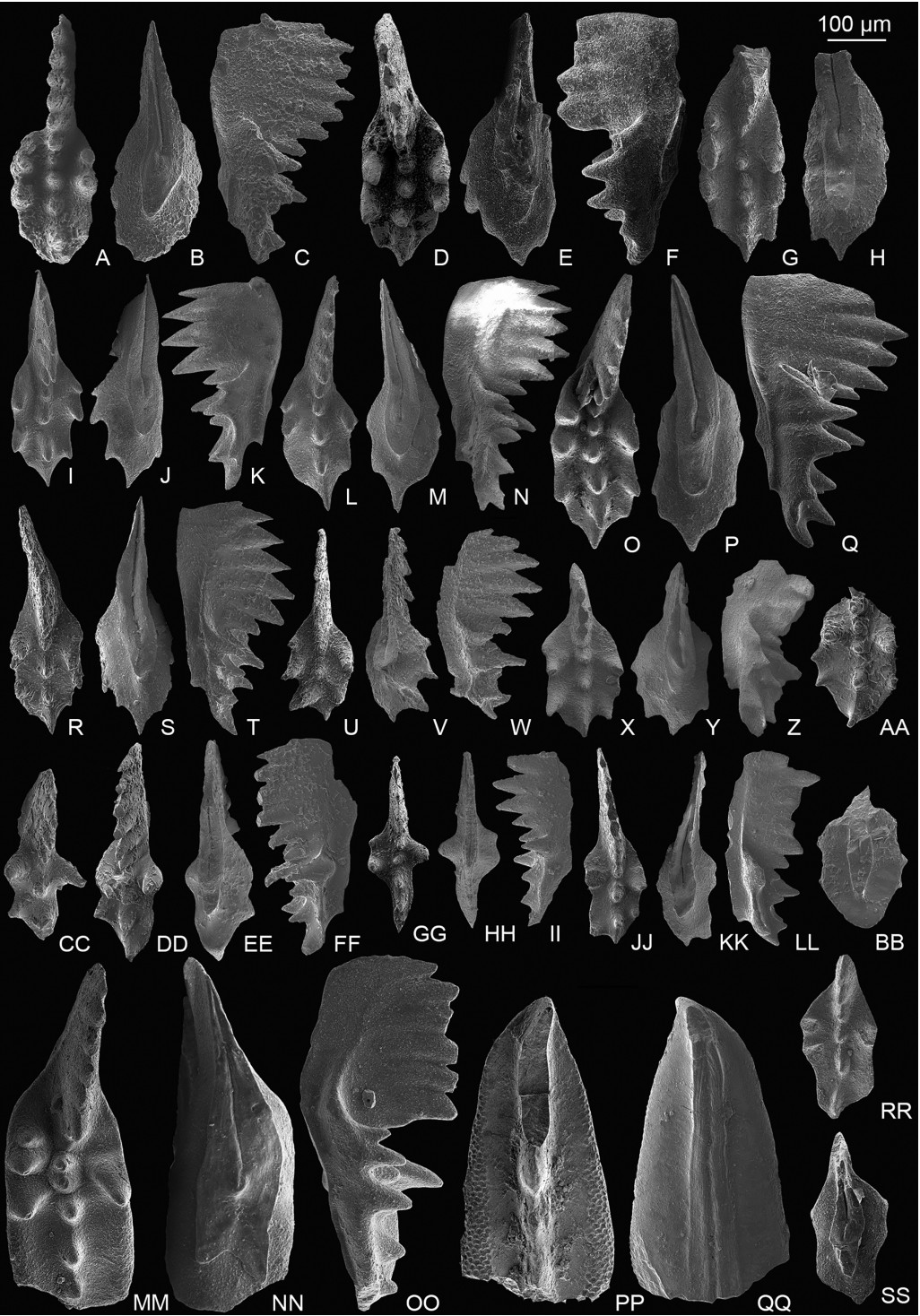

**Figure 3 SEM images of conodonts from the Potou section and the Madoupo section.** (A–T) *Epigondolella carinata* Orchard, 1991b; A–C, D–F, I–K, L–N, O–Q catalog numbers are MDP7-038, MDP8-025, MDP7-113, MDP7_i117and MDP10-059, respectively, all from the Nanshuba Formation; G–H, R–T, catalog numbers are POT1_i105 and POT1_i022, respectively, both from the Dashuitang Formation. (U–BB) *Epigondolella* sp.; U–W, POT1_i011, from the Dashuitang Formation; X–Z, MDP8_i059, from the Nanshuba Formation; AA–BB, POT9_i050, from the Nanshuba Formation. (CC–FF, JJ–LL) *Mockina* sp.; CC, JJ–LL, catalog numbers are POT2-57 and POT2_i058, respectively, all

**Figure 3** (continued)
from the Dashuitang Formation; DD–FF, POT6-32, from the Nanshuba Formation. (GG–II) ?*Mockina bidentata* (*Mosher, 1968*), POT9-48, from the Dashuitang Formation. (MM–OO) ?*Ancyrogondolella praeslovakensis* (Kozur, Masset & Moix, 2007), MDP8-001, from the Nanshuba Formation. (PP–QQ) *Norigondolella?* sp. indet., POT6_i033, from the Nanshuba Formation. (RR–SS) *Mockina zapfei* (*Kozur, 1973*), a juvenile specimen, MDP8-2204, from the Nanshuba Formation. MDP and POT is the abbreviation of Madoupo and Potou, respectively, indicating the specimens come from the two sections respectively.               

expanded and moderately excavated basal cavity; and these features are apparently different from the morphological characteristics of the existing species of *Zieglericonus* which has only one conical cusp, widely expanded and deeply excavated base. It is easy to confuse the two conic elements with some broken ramiform elements of multi-element apparatuses. Many studies show that the groove or basal cavity extends from either side of the pit under the cusp in $S_0$, $S_2$, $S_{3-4}$, and M elements of a multi-element apparatus (*Goudemand et al., 2011*, *2012*; *Orchard, 2005*; *Zhang et al., 2017*; *Demo, 2017*; *Huang et al., 2019a*, *2019b*; *Zeng et al., 2021*), which means that if the four types of ramiform elements broke around the cusp, there should be fracture marks on both ends of the basal groove. However, the well-preserved anterior end of the basal cavity in both conic elements from the Potou section and the Madoupo section shows no interruption, and hence can't be the broken $S_0$, $S_2$, $S_{3-4}$, and M elements. The two conic elements also can't be broken grodelliform $S_1$ elements as there is no additional node develops before the terminal cusp or the denticle (commonly there are two denticles) after the terminal cusp is long. In view of the small sample size, the two conic elements are temporarily classified as ?*Zieglericonus* sp.

*Mockina medionorica* (Figs. 5Y–5BB), which has previously been reported only in the Alaunian of the middle Norian, was discovered by us in the Sevatian strata. The occurrence of *M. medionorica* and *M. elongata* (Figs. 5A–5X) in the *M. bidentata* Zone may indicate that many conodont species considered as middle Norian survived into the late Norian.

Transitional forms between *M. bidentata* and *P. andrusovi* occur in the Potou section. Large *M. bidentata* with long blades occur from levels 14 to 19 m in the Potou section (from POT4 to POT8), below the interval with small transitional forms with no more than six denticles (Figs. 6VV–6AAA) and above the interval having large transitional forms with more than seven denticles (Figs. 6PP–6QQ, 6RR–6SS). Therefore, above the sampling layer of POT8 in the Potou section, the entire evolutionary line from the large *M. bidentata* to *P. andrusovi* is present in stratigraphic order, which is *M. bidentata* –> transitional form between *M. bidentata* and *P. andrusovi* –> *Parvigondolella* sp. –> *P. andrusovi* (shown by the white arrows in Fig. 6). Therefore, it can be inferred that the age of the Potou section ranges from the upper *M. bidentata* Zone to the lower *P. andrusovi* Zone. The transitional form of $P_1$ elements with only one marginal denticle is also common in the *P. andrusovi* Zone of the Xiquelin section in the same area (*Zeng et al., 2021*, figs. 5.2h, 5.7), thereby indicating a wide distribution in the Sevatian of the Baoshan block.

Another transitional $P_1$ element from *M. bidentata* recovered from the Madoupo section (Figs. 6BBB–6CCC) differs from the transitional form between *M. bidentata* and *P. andrusovi* by the terminal big cusp and the moderately opened and excavated posterior base. *P. lata*, *P. ciarapicae* and *Mi. hernsteini* all have big terminal cusps, but the posterior

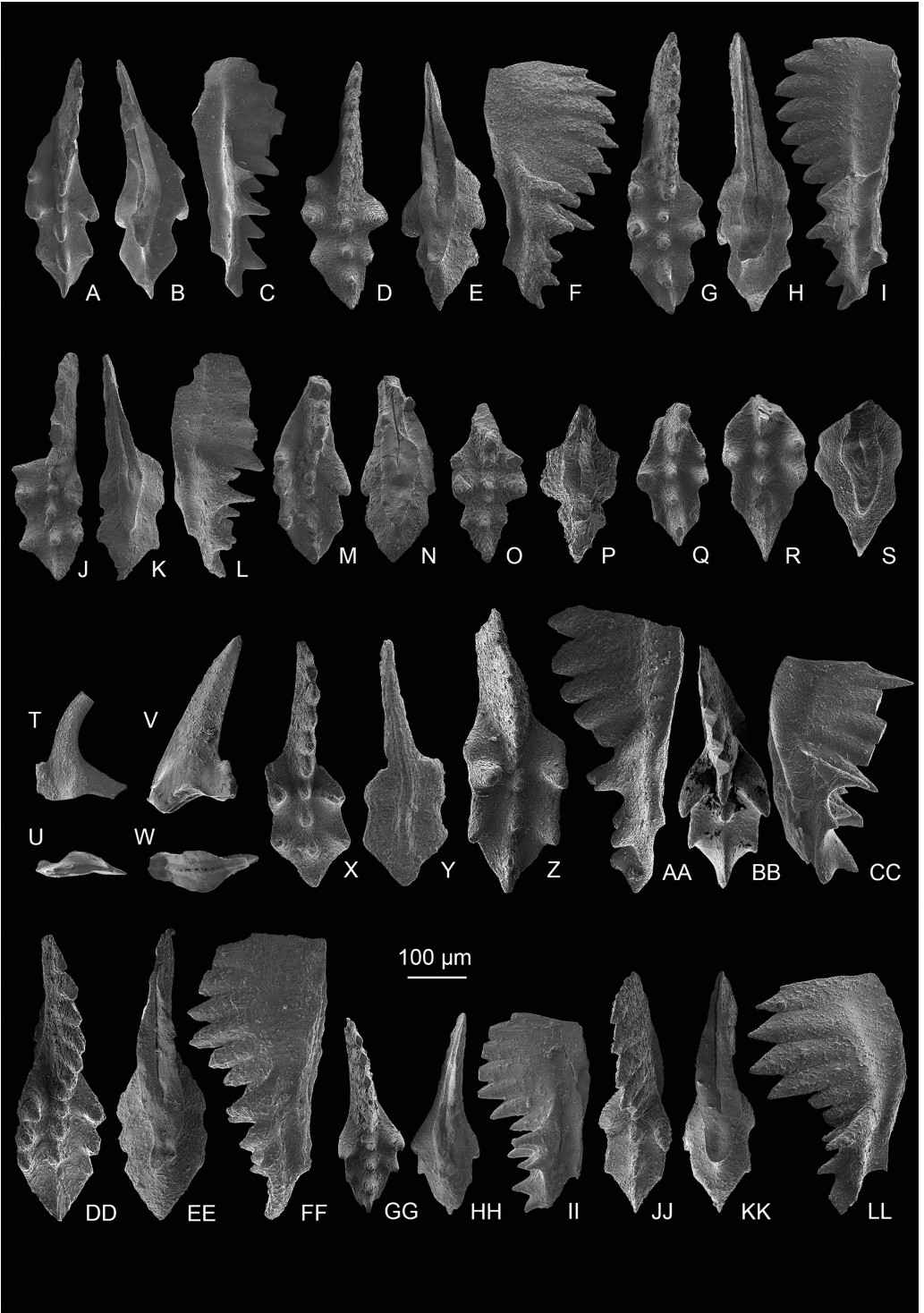

**Figure 4 SEM images of conodonts from the Potou section and the Madoupo section.** (A–S) *Epi-gondolella passerii* (Rigo & Du, 2022); A–C, POT7_i100, from the Nanshuba Formation of the Potou section; D–F, G–I, J–L, M–N, O–P, catalog numbers are POT1_i104, POT2_i097, POT1_i102, POT5_i098 and POT1_i014, respectively, all from the Dashuitang Formation of the Potou section; Q, R–S, catalog numbers are MDP7-3503 and MDP7-3301, respectively, from the Nanshuba Formation of the Madoupo section. (T–W) *Zieglericonus*? sp., from the Nanshuba Formation; T–U, MDP7-110, from the Madoupo section; V–W, POT6_i034, from the Potou section. (X–LL) *Epigondolella* aff. *englandi*

**Figure 4** (continued)
*Orchard, 1991b*; X–Y, Z–AA, DD–FF and JJ–LL, catalog numbers are POT1_i008, POT1_i020, POT1_i060 and POT1_i009, respectively, all from the Dashuitang Formation of the Potou section; BB–CC, MDP10-061, from the Nanshuba Formation of the Madoupo section; GG–II, POT8_i036, from the Nanshuba Formation of the Potou section.

base or keel (commonly posterior to the pit) of *P. lata* and *P. ciarapicae* is commonly not excavated and hence the pit can be easily discerned. The posterior basal cavity of this $P_1$ element resembles that of *Mi. hernsteini*, therefore it seems that the $P_1$ element can evolve into *Mi. hernsteini* by further widening the groove anterior to the pit and missing the marginal denticle. Therefore, it can be inferred that this $P_1$ element may be a transitional form between *M. bidentata* and *Mi. hernsteini*.

Because the first occurrence level of transitional forms from *M. bidentata* are at the uppermost level of the Madoupo section and the lowermost portion of Potou section, it can be inferred that the layers of the Madoupo section that yielded the most conodonts (MDP7 and MDP8) are older than the sampled lower portion of the Potou section.

*Parvigondolella andrusovi* **Zone**—Lower limit: FO of *P. andrusovi*. Upper limit: FO of *Mi. hernsteini* which was not found in the Potou section and the Madoupo section.

Associated taxa: *P.? vrielyncki* and *M. bidentata*. Below the FO of *P. andrusovi* that defines the base of this zone, a transitional form between *M. bidentata* and *P. andrusovi* (Figs. 6PP–6QQ) occurs in sample level POT11, followed by a specimen that has lost all marginal denticles (*Parvigondolella*. sp., Figs. 6RR–6SS) in sample level POT12.

In China, for a long time, *P. andrusovi* had been reported only in the Xikeng section of the Nadanhada Terrane of northeastern China (*Wang, Kang & Zhang, 1986*, see Table 1). Then, *Dong & Wang (2006)* presented the lateral view of a broken *P. andrusovi* from the Nanshuba Formation of the Dabaozi section in Baoshan area. *Zeng et al. (2021)* displayed another specimen of *P. andrusovi* from the Nanshuba Formation of the Xiquelin section in the Baoshan area. The Poutou section is the third section in the Baoshan area that yields *P. andrusovi*, thereby indicating a wide distribution of this species in the Sevatian of the Baoshan area.

## Systematic paleontology

Class Conodonta *Pander, 1856*
Order Ozarkodinida *Dzik, 1976*
Superfamily Gondolelloidea *Lindström, 1970*
Family Gondolellidae *Lindström, 1970*
Genus *Epigondolella Mosher, 1968*

*Type species*—*Polygnathus abneptis Huckriede, 1958* from the Alaunian (*Cyrtopleurites bicrenatus* Zone) of Sommeraukogel, Austria.

*Remarks*—Genus *Epigondolella* is characterized by strongly denticulate anterior and posterior lateral platform margins, by a high blade with most part or even the whole free from the platform, by anteriorly located or rarely sub-centrally positioned cusp and pit,

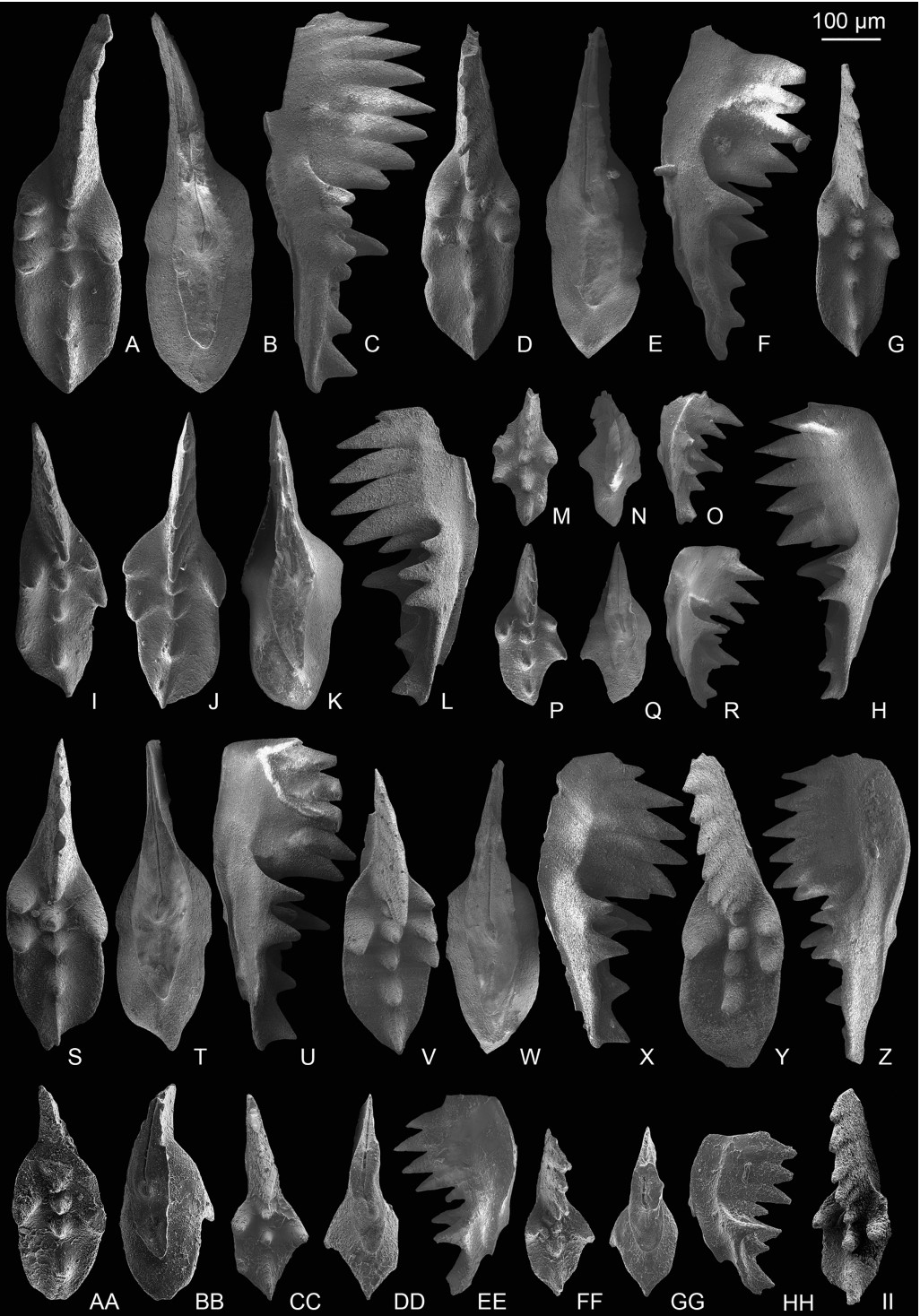

**Figure 5 SEM images of conodonts from the Nanshuba Formation of the Madoupo section.** (A–X) *Mockina elongata* (*Orchard, 1991b*); A–C, D–F, G–H, I, J–L, M–N, P–R, S–U, V–X, catalog numbers are MDP8_i053, MDP8_i060, MDP8_i058, MDP8_i036, MDP8_i045, MDP8_i051 (early juvenile specimen), MDP8_i049 (late juvenile specimen), MDP8_i048 and MDP8_i041, respectively. (Y–BB) *Mockina medionorica Kozur, 2003*; Y–Z, AA–BB, catalog numbers are MDP12-i118 and MDP8-i039, respectively. (CC–EE) *Epigondolella* aff. *englandi Orchard, 1991b*, a juvenile specimen, MDP7-112. (FF–II) *Mockina bidentata* (*Mosher, 1968*); FF–HH, II, catalog numbers are MDP7-3902 and MDP10-060, respectively.

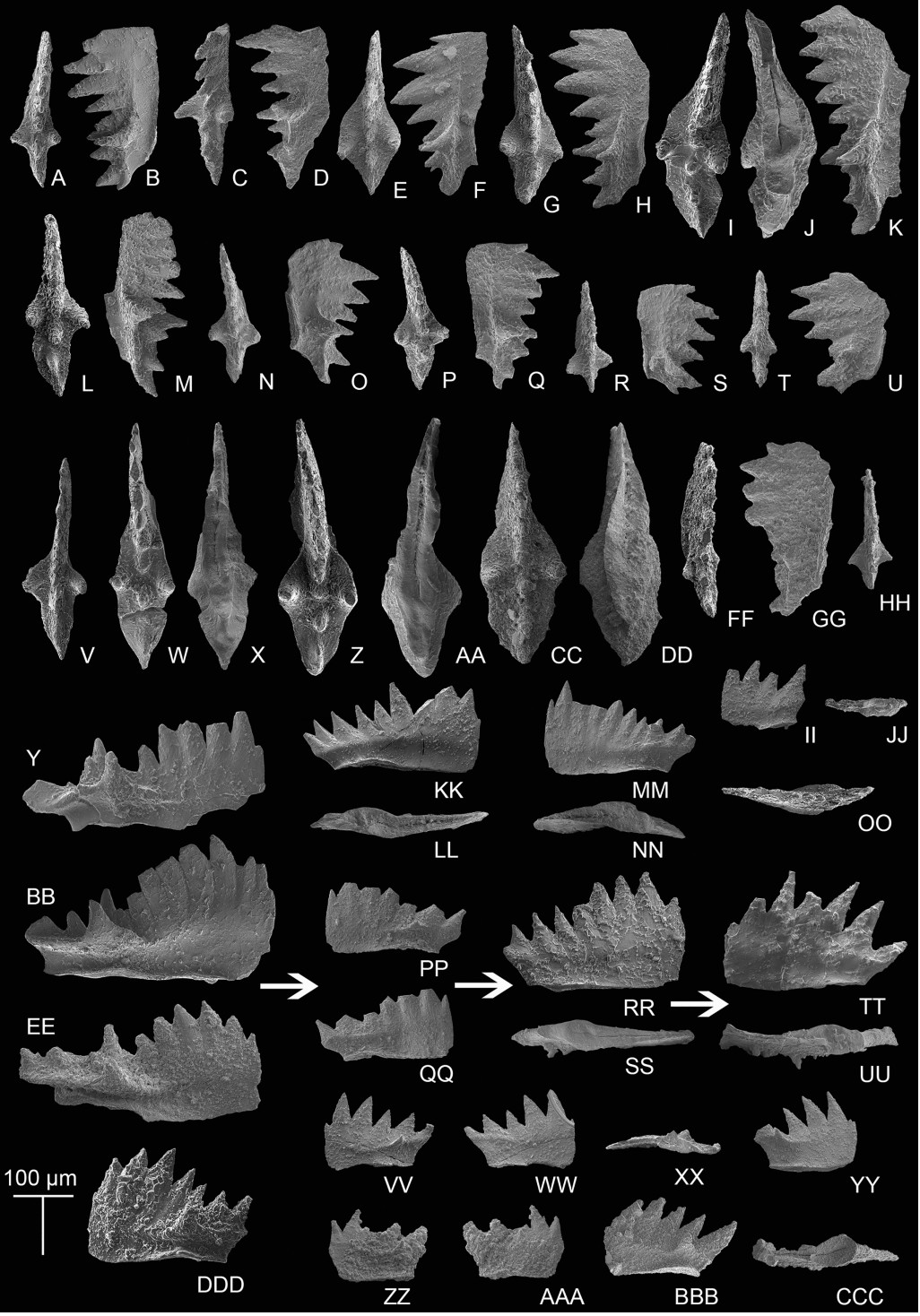

**Figure 6 SEM images of conodonts from the Potou section and the Madoupo section.** (A–JJ, DDD) *Mockina bidentata* (*Mosher, 1968*); A–B, C–E, E–F, G–H, I–K, L–M, N–O, P–Q, R–S, T–U, V, II–JJ, catalog numbers are POT1_i006, POT1_i007, POT1_i018, POT1_i023, POT1_i013, POT1_i015, POT1_i016, POT1_i021, POT1_i017, POT1_i019, POT3-1 and POT1_i024, respectively, all from the Dashuitang Formation; W–Y, Z–BB, CC–EE, FF–GG, HH, DDD, catalog numbers are POT4_i030, POT5_i047, POT8_i038, POT8_i040, POT10_i081 and POT14-1, respectively, all from the Nanshuba Formation; II–JJ, early juvenile specimen. (KK–LL, MM–OO) *Pavigondolella? vrielyncki Kozur & Mock, 1991*,

**Figure 6 (continued)**
catalog numbers are POT13_i061 (from the Nanshuba Formation) and POT1_i059 (from the Dashuitang Formation), respectively. (PP–QQ, VV–AAA) transitional form between *M. bidentata* and *P. andrusovi*; PP–QQ, POT11_i082, from the Nanshuba Formation; VV–XX, YY, ZZ–AAA, may be juvenile P$_1$ elements, catalog numbers are POT1_i010, POT1_i103 and POT8_i089, respectively; VV–YY, from the Dashuitang Formation; ZZ–AAA, from the Nanshuba Formation. (RR–SS) *Parvigondolella*. sp., POT12_i054, from the Nanshuba Formation. (TT–UU) *Parvigondolella andrusovi* Kozur & Mock, 1972, POT13_i057, from the Nanshuba Formation. (BBB–CCC) ?transitional form between *M. bidentata* and *Mi. hernsteini*, MDP25-1, from the Nanshuba Formation of the Madoupo section. A–AAA, from the Potou section. White arrows show the evolutionary trend from *M. bidentata* to *P. andrusovi*.

and by a single keel with either a pointed, rounded, blunt, obliquely truncated, or squared or sinuous keel end which may bear a vestige of the secondary keel on one side.

The characteristics of the base of Norian conodonts are considered to be very important in the taxonomy of the genus (Ishida & Hirsch, 2001; Orchard, 2018) because changes in the bases are considered to reflect the evolution of Late Triassic conodonts (Kozur, 1990; Giordano et al., 2010; Bertinelli et al., 2016; Karádi et al., 2020). Orchard (2018) classified lower Norian ornate P$_1$ elements with a bifid keel into *Ancyrogondolella*, and mid-Norian ornate and asymmetric or less commonly symmetric P$_1$ elements with a single keel (pointed, or squared-off, or obliquely truncated or sinuous) into *Epigondolella*, which reconstructs the phylogeny between genera *Ancyrogondolella* and *Epigondolella*. In the light of the fact that bifid-keeled ornate P$_1$ elements were mostly substituted by single-keeled ornate P$_1$ elements from the early Norian to the late Norian and by the abundant single-keeled ornate P$_1$ elements from the upper Norian of Baoshan area (Potou, Madoupo and the Xiquelin sections in Zeng et al. (2021)), we follow the definition of *Epigondolella* as revised by Orchard (2018). But the symmetry of the posterior platform was not differentiated in genus *Epigondolella* and there might be intraspecific differences.

*Epigondolella carinata* Orchard, 1991b
Figures 3A–3T
1983 *Epigondolella postera* (Kozur and Mostler) population; Orchard, p. 186–188, figs. 11A, C.
1991b *Epigondolella carinata* n. sp.; Orchard, p. 308, pl. 5, figs. 4, 5, 10.
2007 *Epigondolella carinata* Orchard; Carter & Orchard, pl. 2, figs. 15, 21.
?2007b *Epigondolella carinata*; Orchard et al., figs. 8.14–8.15.
2020 *Mockina carinata*; Du et al., figs. 3.1–3.2, 3.5.
2020 *Mockina* sp.; Du et al., figs. 3.10–3.11.

*Materials*—Five P$_1$ elements from MDP7, two P$_1$ elements from MDP8, two P$_1$ elements from MDP10, two P$_1$ elements from POT1.

*Description*—The P$_1$ elements have an ovoid platform bearing two and one high denticles respectively on each anterior lateral margin. The posterior marginal denticles increase in number as the element posteriorly grows longer. The length ratio of the platform-to-element is about four sevenths. The blade is as long as half of the element and most of it is

free from the platform. The posterior-most and anterior-most denticles of the blade are commonly smaller and lower than other denticles, which are of similar height; therefore the blade gradually transitions to the low carina, which consists of three to four discrete nodes. There are two to three carinal nodes behind the anteriorly located cusp. The posterior carinal nodes may extend beyond or near the pointed platform end, and increase in height and inclination after the cusp. The pit is anteriorly migrated and is located on a medium wide keel, which is posteriorly prolonged. The keel end is pointed to narrowly rounded.

*Comparison*—*Epigondolella passerii* has a platform constriction. *E. englandi* and *E.* aff. *englandi* have only one high marginal denticle on each platform side. *M. postera* has an asymmetrical posterior platform and a posterior carina that never reaches to the platform end. *M. medionorica* has smooth posterior platform margins and a short posterior carina that always stops before the platform end.

*Remarks*—The $P_1$ elements resemble *E. carinata* in the marginal denticulation and platform shape, and only differs in having a larger size and longer blade and free blade. The longer blade and free blade may be intraspecific difference. This species, for a long time, was only reported in North America. The occurrence of this species in the Hongyan (*Du et al., 2020*), Potou and Madoupo sections on the Baoshan Block indicates that it is also present in Sevatian strata of eastern Tethys and hence is likely a more globally distributed species.

*Occurrence*—Sevatian (*M. bidentata* Zone) in the Nanshuba Formation of the Madoupo section and in the Dashuitang Formation of the Potou section, China (this study). Sevatian in the Nanshuba Formation of the Hongyan section (*Du et al., 2020*). Sevatian to early Rhaetian? at Kennecott Point on Queen Charlotte Islands, Canada (*Carter & Orchard, 2007*). Middle Alaunian in the Pardonet Formation of Pardonet Hill, British Columbia, Canada (*Orchard, 1991b*).

*Stratigraphic range*—Middle Alaunian to Sevatian.

*Epigondolella* aff. *englandi* Orchard, 1991b

Figures 4X–4LL, 5CC–5EE
2019 *Mockina englandi* (*Orchard, 1991a*); Wang et al., p. 87, figs. 6.2–6.3.

*Materials*—Five specimens form POT1, one specimen from POT8, two specimens from MDP7, one specimen from MDP10.

*Description*—The $P_1$ elements are characterized by a long free blade and a relatively short platform, as well as the symmetrically arranged marginal denticles. The platform is commonly ovoid and sub-symmetrical. The posterior platform is pointed. The anterior-most pair of marginal denticles are the highest, and the following pairs of marginal denticles decrease in height and are more posteriorly inclined and outwardly projected. The blade is very high and comprised of six to seven highly fused denticles which are low at

both ends and hence gradually transition to the low carina on the platform. The free blade is nearly as long as half of the element. The carinal nodes are discrete and extend to or within the posterior platform edge. There are two to three carinal nodes behind the anteriorly located cusp. The anterior groove beneath the blade is widely opened. The pit is strongly anteriorly shifted. The keel end is commonly pointed and posteriorly prolonged, and rarely keeps remnant secondary keels which are weakly and asymmetrically bifurcated. The form with a weakly bifid keel end bears three pair of marginal denticles, which may represent an intermediate form between an ancestor and the *E.* aff. *englandi*.

*Comparison*—*Epigondolella carinata* bears two and one anterior marginal denticles respectively on the outer platform and the inner platform. *Ancyrogondolella equalis* has a bifurcated keel end and a rectangular platform, and its marginal denticles are not asymmetrically arranged.

*Remarks*—The $P_1$ elements resemble *E. englandi*, but are distinguished from the latter by the development of a longer blade and free blade, and by having more numerous posterior marginal denticles or a weakly bifurcated keel end; however, these probably have an affinity to those of *E. englandi*. *Wang et al. (2019)* presented two specimens of *E. englandi* which are also from Baoshan block that have longer blade and free blade than the holotype of *E. englandi,*. Therefore, this form of $P_1$ elements is widely distributed in the Sevatian of the Baoshan block; indicating the occurrence may be geographically limited and hence this type of conodont might be a subspecies of *E. englandi*.

*Occurrence*—Sevatian (*M. bidentata* Zone) of the Dashuitang Formation and the Nanshuba Formation, China (in this study). Sevatian of the Dashuitang Formation in the Hongyan Section, Baoshan city, China (*Wang et al., 2019*).

*Stratigraphic range*—Sevatian.

*Epigondolella passerii* (Rigo and Du, 2022)

Figures 4A–4S
1980 *Epigondolella postera* Kozur and Mostler; Krystyn, pl. 13, figs. 17, 18.
?1990 *Epigondolella postera*; Wang and Wang, pl. 1, fig. 10.
2003 *Mockina* cf. *carinata* (Orchard); Channell et al., fig. 3A/12.
2022 *Mockina passerii* n. sp. Rigo and Du; Jin et al., figs. 4.10–4.12.

*Materials*—10 $P_1$ elements.

*Description*—The slender $P_1$ elements have a thin and biconvex platform with a pronounced constriction after a pair of highest anterior marginal denticles. The platform end is pointed. The anterior platform possesses two to three denticles on the outer margin and one to two denticles on the inner margin. The posterior platform is narrower than the anterior platform and bears a pair of unevenly developed and outwardly projected denticles on each margin. The blade is high and longer than half of the element, consisting of seven to eight highly fused denticles and hence forming a crest shape and gradually

transitioning to the cusp. The cusp is located on the anterior platform and is generally followed by two to three posteriorly inclined carinal nodes. The posterior carina extends to, beyond or within the platform end. The pit is situated under the anterior platform in a narrow keel. The keel end is posteriorly prolonged and pointed.

*Comparison*—*Epigondolella serrulata* possesses more marginal denticles on the platform and a weaker constriction after a pair of highest anterior marginal denticles. The anterior platform of *E. englandi* possesses only one pair of unevenly developed marginal denticles on each side. *E.* aff. *englandi* has sub-symmetrically arranged marginal denticles and an oval platform with no constriction. *E. carinata* has no or very weaker constriction on the middle part of the platform.

*Occurrence*—Lower Sevatian (*M. bidentata* Zone, this study) in the Dashuitang Formation and the Nanshuba Formation of Potou section, Baoshan city, China. Sevatian at Sommeraukogel of Austria (*Krystyn, 1980*). Sevatian in the Dashuitang Formation of Hongyan-B section, Baoshan city, SW China (*Jin et al., 2022*).

*Stratigraphic range*—Sevatian.

Genus *Mockina* Kozur, 1990

*Type species*—*Tardogondolella abneptis postera* Kozur & Mostler, 1971 from the middle Norian of Sommeraukogel, Austria.

*Remarks*—Compared with *Epigondolella*, *Mockina* is characterized by nearly smooth lateral posterior platform margins which may sparsely develop small nodes.

*Mockina bidentata* (*Mosher, 1968*).

Figures 5FF–5II, 6A–6JJ, 6DDD, 7U–7CC
1958 *Polygnathus abneptis* n sp.; Huckriede, p. 156, pl. 14, figs. 32, 58.
1968a *Epigondolella bidentata* n. sp.; Mosher, p. 936, pl. 118, figs. 31–35.
1972 *Metapolygnathus bidentatus* (Mosher); Kozur, pl. 7, figs. 3–9.
1972 *Epigondolella bidentata* Mosher Kozur & Mostler, pl. 4, figs. 3–5.
1980 *Metapolygnathus bidentatus*, Kovács & Kozur, pl. 15, fig. 1.
1980 *Epigondolella bidentata* Mosher, Krystyn, pl. 14, figs. 1–3.
1983 *Epigondolella bidentata* population, Orchard, figs. 14 O–Q, S, W, X, figs. 15W, X.
1984 *Epigondolella bidentata*; Meek 1984, pl. 1, figs. 1–4.
1985 *Epigondolella bidentata*; Wang & Dong, p. 127–128, pl. 1, figs. 1–3, 26.
1991b *Epigondolella bidentata*; Orchard, p. 307–308, pl. 4, fig. 12.
2003 *Mockina bidentata* (Mosher); Channell et al., pl. A2, figs. 44, 46–48, 51, 54; pl. A3, figs. 3, 4, 6, 7, 9, 25, 27, 28, 37, 39, 41, 42, 47, 48, 50, 54, 56, 71, 72, 74–79.
2005 *Epigondolella bidentata*; Bertinelli et al., fig. 4/5.
2005 *Epigondolella bidentata*; Rigo et al., fig. 4/6.
2005 *Epigondolella* ex gr. *bidentata* Orchard; Hornung, p. 111, pl. 1, fig. e.
2007 *Mockina bidentata*; Moix et al., p. 294, pl. 2, figs. 2, 3.

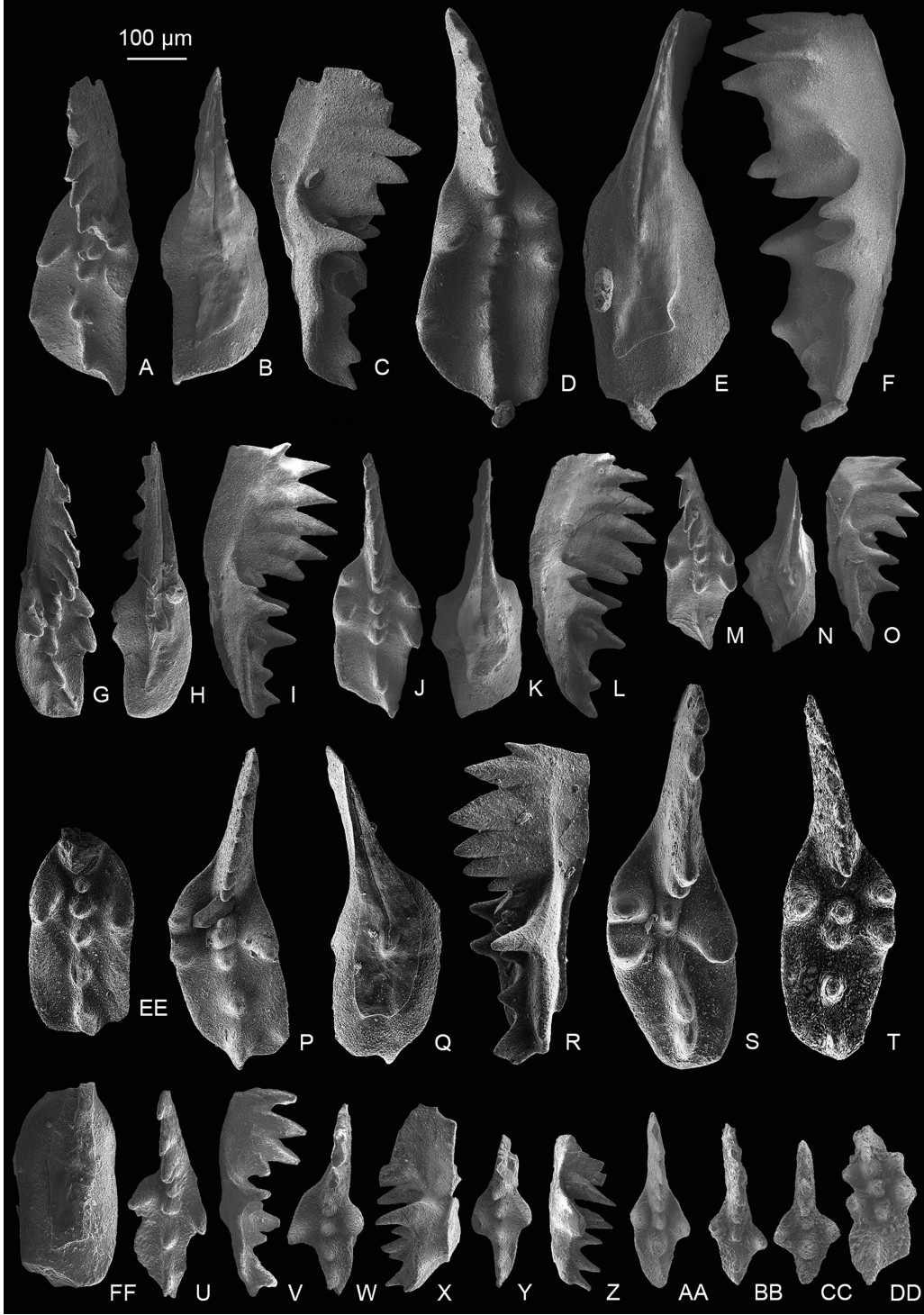

**Figure 7 SEM images of conodonts from the Nanshuba Formation of the Madoupo section.**
(A–O, DD) *Mockina zapfei* (*Kozur, 1973*); A–C, D–F, G–I, J–L, M–O, DD, catalog numbers are
MDP8_i040, MDP8_i061, MDP8_i109, MDP8_i039, MDP8_i043 and MDP10-4402, respectively. (P–T)
*Mockina* sp. A; P–R, MDP8-031; S, MDP8-003; T, MDP8-032. (U–CC) *Mockina bidentata* (*Mosher,
1968*); U–V, W–X, Y–Z, AA, BB, CC, catalog numbers are MDP8_i044, MDP7_i115, MDP7_i116,
MDP10-4503, MDP10-5207 and MDP7-3203, respectively. (EE–FF) ?*Mockina medionorica Kozur, 2003*,
MDP8-011.                               

2007 *Epigondolella bidentata*; Carter & Orchard, pl. 2, figs. 14, 22–25.
2007a *Epigondolella bidentata*; Orchard et al., figs. 8.10, 8.11, 8.13, 8.17–8.20.
2007b *Epigondolella bidentata*; Orchard et al., pl. 1, fig. 22.
2007 *Epigondolella bidentata*; Krystyn et al., pl. 1, figs. 5, 6; non pl. 1, figs. 7–14.
2009 *Epigondolella bidentata*; Rožič et al., fig. 9e.
2010 *Mockina bidentata*; Giordano et al., figs. 3/1, 3/2.
2012 *Mockina bidentata*; Mazza et al., p. 120, pl. 7, fig. 7.
2012 *Epigondolella bidentata*; Gallet et al., fig. 3.1.
2016 *Mockina bidentata*; Rigo et al., fig. 3/3.
2016 *Mockina bidentata*; *Karádi, Pelikán & Haas, 2016*, pl. 1, fig. 8; pl. 4, fig. 4.
2018 *Mockina bidentata*; Yamashita et al., p. 183, figs. 8.4, 8.5.
2020 *Mockina bidentata*; Karádi et al., fig. 5 S.
2020 *Mockina bidentata*; Du et al., figs. 3.6, 3.15.
2021 *Mockina bidentata*; Du et al., figs. 3.1, 3.2.
2021 *Mockina bidentata*; Zeng et al., figs. 1/b, 2/2b, 3/1n–1o, 4/3d.
2022 *Mockina bidentata*; Jin et al., figs. 4.13–4.15.

*Materials*—31 $P_1$ elements.

Description —Platform is short and reduced and varies in width. One pair of denticles are located on the anterior platform margins, after which the platform changes from extremely reduced to moderately reduced. The pair of marginal denticles are either both developed on two sides of the unit or are both absent in the early juvenile element. The blade varies in length and consists of four to nine denticles, which are fused to different degrees.
The denticles of the blade may be high in the middle and lower on both ends or high on the anterior and gradually descending posteriorly. The cusp is located in a position parallel with the line of marginal denticles and is followed by one to three additional carinal nodes. The pit is situated beneath the cusp. The keel end is prolonged relative to the pit and pointed. In lateral view, the basal edge is sub-straight or is slightly concave beneath the transition between the blade and the platform. Two specimens from Madoupo section (Figs. 5FF–5II) have a convex basal edge in lateral view and a broad platform.

*Remarks*—Specimens of various *M. bidentata* present a possible evolutionary trend for this species from the early or middle Sevatian to the late Sevatian. On the basis of different features of the carina in *M. bidentata*, *Moix et al. (2007)* discerned two morphotypes of *M. bidentata*, and these are also present in *M. bidentata* fauna from the Potou section. One morphotype (Figs. 6W–6EE) resembles the holotype with a long anterior blade consisting of many highly fused denticles and a long posterior carina of three nodes. The highest occurrence of this morphotype (POT8) in the Potou section is below the FO of *P. andrusovi*, which contrasts with its re-appearance within *Mi. hernsteini-P. andrusovi* Zone observed by *Moix et al. (2007)*. The second, somewhat smaller, morphotype with a shorter blade consisting of less fused and few relatively big and wide denticles is below both the occurrences of the other morphotype of *M. bidentata* and of the *M. bidentata/ P. andrusovi* transitional form and *P. andrusovi* in several Tethyan sections and on the

Tavuşçayırı Block (*Moix et al., 2007*). The occurrence level of this second morphotype of *M. bidentata* in the Potou section (Figs. 6A–6H, 6L–6V) is similar to the inferred level below the *Mi. hernsteini-P. andrusovi* Zone. *M. bidentata* increases in size from the sample layer 1 (POT1) to sample layer 8 (POT8), then reduces its size just below the occurrence of *M. bidentata/P. andrusovi* transitional forms (from sampling layer POT11 to POT13).

At present, *M. bidentata* has been widely reported in the Tethys (*Channell et al., 2003*; *Krystyn et al., 2007*; *Moix et al., 2007*; *Giordano et al., 2010*; *Mazza, Rigo & Gullo, 2012*; *Karádi et al., 2020*; *Du et al., 2020, 2021*; *Zeng et al., 2021*; *Jin et al., 2022*), in North America (*Mosher, 1968*; *Orchard, 1983, 1991b*; *Orchard et al., 2007a, 2007b*; *Carter & Orchard, 2007*) and in Japan (*Yamashita et al., 2018*). All the illustrated specimens from these publications display a narrow platform and straight or slightly concave basal edges in lateral view, which conform to the holotype. It is noteworthy that three bidentate specimens from the Madoupo section differ from the bidentate $P_1$ elements from the Potou section and from other *M. bidentata* in the Madoupo section by having distinct convex rather than sub-straight basal edges in lateral view and possessing broader platforms (Figs. 5FF–5II); but more data are needed to confirm whether the different morphologies are intraspecific.

*Stratigraphic range*—from lowermost Sevatian (*M. bidentata* Zone) to (middle?, *Mi. ultima* Zone) Rhaetian.

*Mockina elongata* (*Orchard, 1991b*)

Figures 5A–5X
1991b *Epigondolella elongata* n. sp.; Orchard, p. 308, pl. 4, figs. 4–6, 15, 20, 21.
2005 *Epigondolella elongata* Orchard, 1991a; Rigo et al., fig. 4.4.
2018 *Mockina elongata* (*Orchard, 1991b*); Yamashita et al., p. 183, figs. 8.6, 8.7.
2022 *Mockina mosheri* morphotype B; Jin et al., fig. 4.8.

*Materials*—28 $P_1$ elements from MDP8.

*Description*—The platform after the anterior marginal denticles gradually tapers to the pointed platform end, to form a long elliptical outline. The smooth posterior platform is relatively wider on the inner sider than the outer side, but both margins are convex in outline. The length ratio of the platform to the element is about three fifths. The anterior platform bears two or three high denticles on one margin and one higher denticle on the other. The blade consists of six to seven high denticles with the basal part fused and the upper part discrete. The anterior-most or posterior-most denticle of the blade may be relatively smaller or lower, and thus the blade moderately or abruptly decreases to the low carinal nodes on the platform. The low carina consists of four to five discrete posteriorly inclined nodes, commonly extends beyond the platform end or aligns with the terminal marginal denticle, and rarely has the posterior two carinal nodes fused with the terminal marginal denticle. The cusp is usually higher than the adjacent carinal nodes and is followed by three to four carinal nodes with the last one being the largest. The cusp located on the anterior platform is approximately aligned with the middle position of the two to

three marginal denticles. The keel extends along the platform and terminates at three fifths along the posterior platform. The keel after the pit is straight to slightly curved. The keel end is pointed or lanceolate and prolonged far from the pit. The pit is located below the cusp at about the anterior third of the platform.

The juvenile forms are much smaller in size and have the same ratio of platform length to the unit length. The carina of juvenile specimens consists of four high denticles (blade) and four lower carinal nodes. The keel grows along the platform, and both have pointed ends. The pit in the juvenile form is located beneath the anterior platform and surrounded by a bulged loop.

*Comparison*—*Mockina postera* has a shorter platform, an apparent asymmetrical posterior platform and a shorter posterior carina that never reaches the platform end. *M. matthewi* has at least two anterior platform marginal denticles and a broader platform. *M. medionorica* has a smooth platform end and a shorter posterior carina that also never reaches the platform end. *Orchardella*? *multidentata* has more anterior marginal denticles and a stronger posterior carina.

*Remarks*—All $P_1$ elements have an elongate ellipsoid platform with two to three denticles on one anterior platform margin, with one denticle on the other margin and with an inornate posterior platform. The pit is anteriorly shifted, and prominent carinal nodes usually extend to the posterior platform end or occasionally not. Although the holotype of "*Epigondolella*" (= *Mockina*) *elongata* (*Orchard, 1991b*, pl. 4, figs. 15, 20, 21) has a slenderer, longer platform and more posteriorly elevated carinal nodes than these $P_1$ elements, the morphological features of the $P_1$ elements match well with the description in the "Diagnosis" of *M. elongata* and resemble the paratype (*Orchard, 1991b*, pl. 4, figs. 4–6).

The occurrence of *M. elongata* is global, as indicated by the occurrence of this species in western Tethys (*Rigo et al., 2005*), in eastern Tethys (*Jin et al., 2022*; and this study), in the Panthalassa (*Yamashita et al., 2018*) and in North America (*Orchard, 1991b*).

*Occurrence*—Lower Sevatian in the Madoupo section. Middle Alaunian in the British Columbia, North America (*Orchard, 1991b*). Upper Alaunian in the Sasso di Castalda section of southern Italy (*Rigo et al., 2005*). Lower Sevatian (*M. bidentata* Zone and occurrence within radiolarians zone TR6B (*Tnalatus robustus-Lysemelas olbia*) through TR7 (*Lysemelas olbia*) and disappearance in lower TR8A (*Praemesosaturnalis multidentatus*) in the section Q of Japan (*Yamashita et al., 2018*). Sevatian in the Dashuitang Formation of the Honyan-B section, Baoshan area, western Yunnan, China (*Jin et al., 2022*).

*Stratigraphic range*—from middle Alaunian to lower Sevatian.

*Mockina medionorica* *Kozur, 2003*

Figures 5Y–5BB

1980 *Metapolygnathus multidentatus* (Mosher); Kovács & Kozur, pl. 14, fig. 5.

1987 *Metapolygnathus posterus* (Kozur & Mostler); Vrielynck, p. 157–159, pl. 7, Figs. 10–15.
2001 *Mockina postera* (Kozur & Mostler); Ishida & Hirsch, p. 238, pl. 4, figs. 4, 6.
2002 *Mockina postera*; Hirsch & Ishida, pl. 1, fig. 3.
2003 *Mockina medionorica* n. sp.; Kozur, p. 70, pl. 1, figs. 5, 6.
2003 *Mockina medionorica* Kozur; Channell et al., pl. A3/14–15.
2021 *Mockina medionorica*; Karádi et al., p.16, fig. 7.13.

*Materials*—Two P$_1$ elements.

*Description*—The P$_1$ elements have an oval platform that possesses two and one marginal denticles on each anterior platform side. The platform margins after the anterior marginal denticles are smooth. The blade consists of seven half-fused denticles. Most of the blade is free from the platform. The posterior two denticles of the blade decrease in height toward the posterior and hence gradually descend to the low carina. The low carina consists of four isolated nodes, with the last carinal node being largest and stopping before the smooth platform end. The pit located under the anterior platform. The keel is pronged after the pit. The keel end is pointed. In lateral view, the base is stepped upwards at the transition between the blade and the platform and become sub-straight after the cusp.

*Comparison*—*Mockina postera* has an asymmetrical posterior platform and a pointed platform end. *M. elongata* has a pointed platform end and longer carina after the cusp. *M. slovakensis* differs from it by the blade abruptly descends to the low carina on the platform. *M. matthewi* has two or more anterior marginal denticles on each platform side.

*Remarks*—Kovács & Kozur (1980, pl. 14, fig. 5) presented a specimen of *Metapolygnathus* (*Me.*) *multidentatus*. However, the displayed specimen differs from *Me. multidentatus* by the broader posterior platform, smooth platform end and a shorter posterior carina that never extends to the platform end. It resembles *M. matthewi* by the two pairs of anterior marginal denticles, broad platform, smooth posterior platform margins and the carina, but differs from the latter by the keel end that is narrowly blunt with slight central incision. This kind of keel end was described in the diagnosis of *M. medionorica*. Karádi et al. (2021, p. 16) assigned this specimen to *M. medionorica* and proposed that *M. matthewi* has the broadest platform in the middle. Taking the broadest anterior platform and the centrally incised keel end into consideration, the specimen of *Me. multidentatus* in Kovács & Kozur (1980) conforms more to *M. medionorica*.

*Occurrence*—Lower Sevatian in the Madoupo section. Alaunian in the Rudabánya hills, Hungary (*Kovács & Kozur, 1980*). Upper Alaunian in Cammarata of Sicily, Italy (*Vrielynck, 1987*). Alaunian in the section of Hisaidani, southwestern Japan (*Ishida & Hirsch, 2001*; *Hirsch & Ishida, 2002*). Alaunian of Silická Brezová, Slovakia (*Kozur, 2003*; *Channell et al., 2003*). Lower Alaunian in the Dovško section of Slovenia (*Karádi et al., 2021*).

*Stratigraphic range*—from lower Alaunian to Sevatian.

*Mockina zapfei* (*Kozur, 1973*)

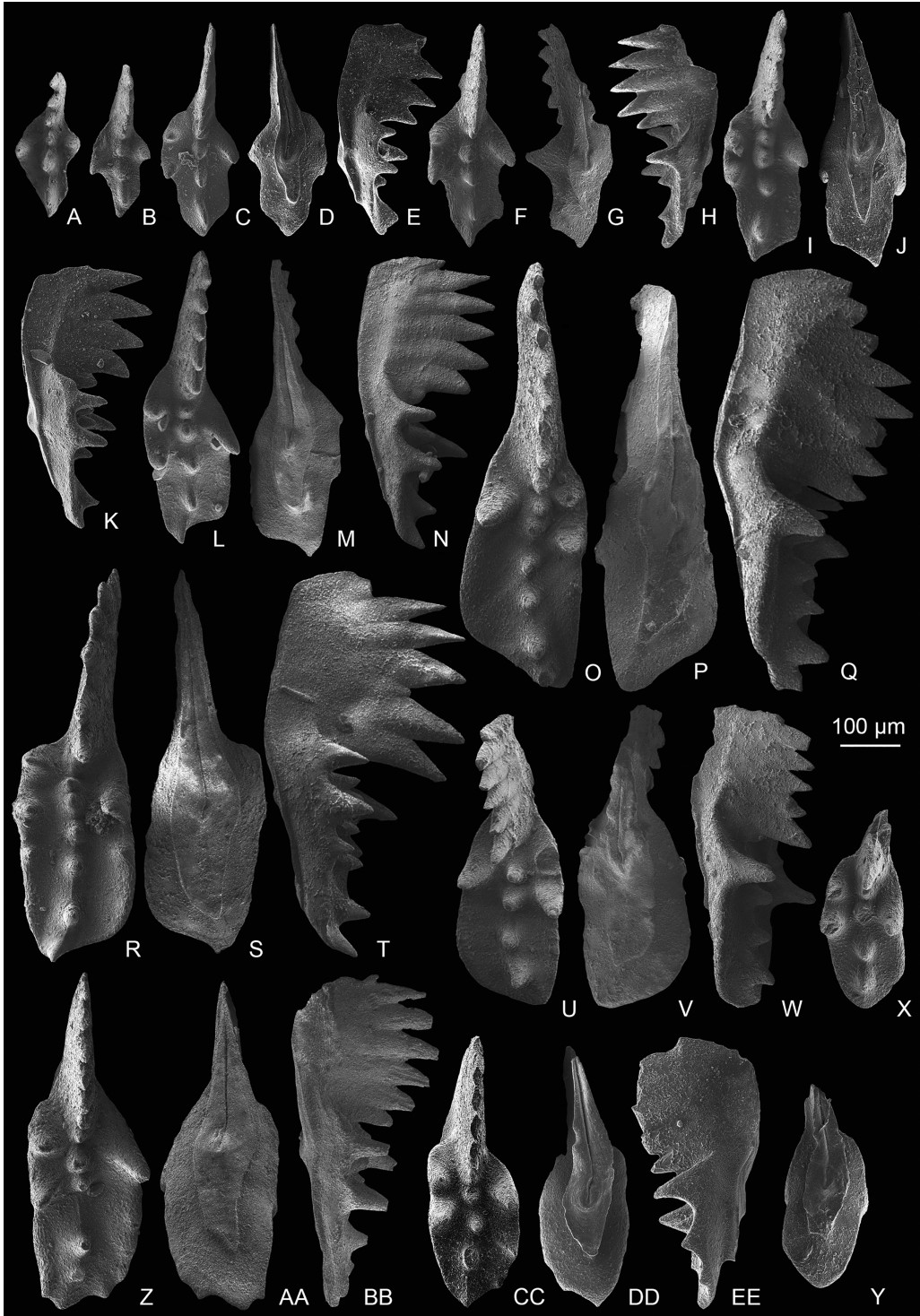

**Figure 8** **SEM images of *Mockina zapfei* (*Kozur, 1973*) from the Nanshuba Formation (Sevatian) of the Madoupo section in Baoshan, western Yunnan, China and its ontogenetic series.** (A–T) Ontogenetic series; A, B, early juvenile species, catalog numbers are MDP8_1802 and MDP8-1901, respectively; C–E, F–H, late juvenile specimens, catalog numbers are MDP8-019 and MDP8_i108, respectively;

**Figure 8** (continued)
I–K, early adult species, MDP8-902; L–N, adult species, MDP8_i035; O–Q, R–T, late adult species, catalog numbers are MDP8_i057 and MDP8_i055, respectively. (U–Y) The platform end of the $P_1$ element develops no terminal denticle; U–W, adult species, MDP_i037; X–Y, late juvenile species, MDP8-2909. (Z–EE) the $P_1$ elements develop small or tiny nodes on the platform end and one of the posterior lateral platform margins; Z–BB, CC–EE, catalog numbers are MDP8_i054 and MDP8-005, respectively.                                                 

Figures 3RR–3SS, 7A–7O, 7DD, 8A–8EE, 9A–9BB
1972 *Metapolygnathus* n. sp.; Kozur, pl.7, fig.1.
1972 *Metapolygnathus* aff. *posterus* (Kozur & Mostler); Kozur, pl. 7, fig. 2.
1973 *Metapolygnathus zapfei* Kozur & Mostler; Kozur, p. 18–20.
1979 *Metapolygnathus zapfei* Kozur; Gaździcki et al., pl. 5, fig. 15.
1983 *Epigondolella postera* population (Kozur & Mostler); Orchard, p. 186, figs. 15 P–R.
1985 *Epigondolella multidentata* Mosher; Wang & Dong, p. 128, pl. 1, figs. 9, 16.
1990 *Epigondolella postera*; Buduro & Sudar, pl. 5, fig. 6–8.
?1991a *Epigondolella postera*; Orchard; pl. 4, figs. 17, 18.
2000 *Epigondolella slovakensis* Kozur & Mock; Martini et al., pl. V, fig. 13, 14.
2003 *Mockina zapfei;* Channell et al., pl. A2, figs. 43, 45, 53, 55; pl. A3, figs. 5, 24, 32, 33, 34, 36, 43, 51, 52, 53.
2005 *Epigondolella* ex gr. *bidentata* Orchard; Hornung, p. 111, pl. 1, fig. B (only).
?2005 *Epigondolella serrulata* Orchard, 1991a; Rigo et al., fig. 4.3.
2021 *Mockina zapfei*; Du et al., fig. 3.12.
2022 *Mockina zapfei*; Jin et al., fig. 4.6.

*Materials*—60 $P_1$ elements from MDP8, one $P_1$ element from MDP10.

*Description*—The $P_1$ elements have a long platform which spans more than half of the entire element, and is always with one side of the posterior platform margin deflecting toward the other side near the platform end. The platform width is from slender to broad, posteriorly elongating during the growth and thus developing a longer posterior platform in adult forms or late forms. The anterior platform bears one to three high denticles on one lateral margin and always one strong and highest denticle on the other. The posterior platform margin is commonly smooth and asymmetric, with the deflected side of the platform wider than the other side. The blade spans nearly half of the element and commonly consists of five to eight highly fused denticles in adult elements, but with most denticles free from the platform. The denticles of the blade are commonly similar in height except the anterior-most one or two denticles which are smaller or lower. The posterior one to two denticles of the blade may decrease in height or size, or commonly in adult or late adult specimens the posterior-most denticle may be abrupt. The cusp is located on the anterior platform and is commonly followed by two or more carinal nodes which increase in height toward the posterior and extend to or within the posterior platform end. The keel is wide around the pit and tapers posteriorly. The keel end is commonly lanceolate in shape and posteriorly prolonged; it deflects toward one side, and terminates at the posterior fourth of the platform. Besides these common morphological features, the abundant

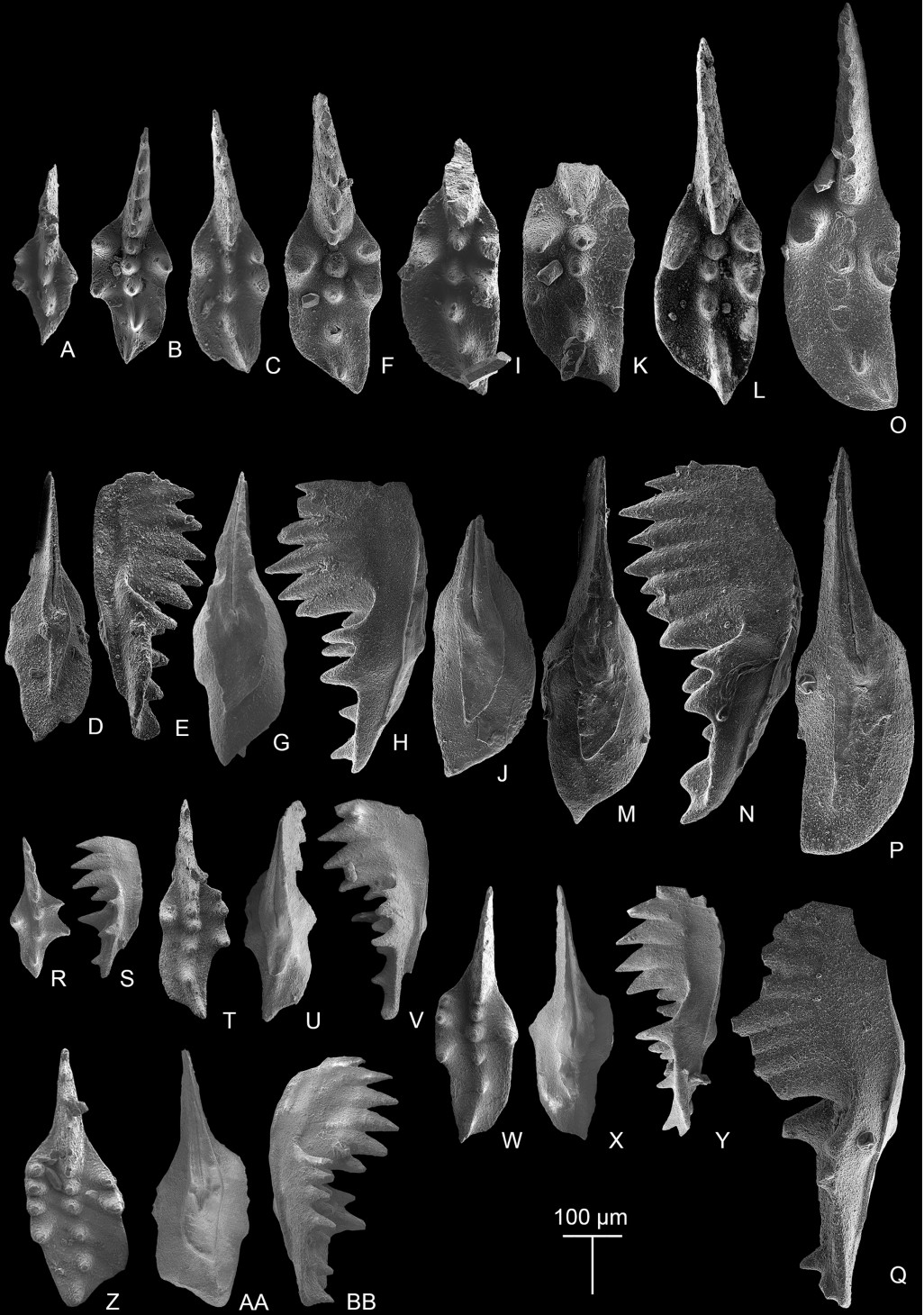

**Figure 9  SEM images of *Mockina zapfei* (*Kozur, 1973*) from the Nanshuba Formation (Sevatian) of the Madoupo section in Baoshan, western Yunnan, China and its ontogenetic series.** (A–N) Ontogenetic series; A, juvenile species, catalog no. is MDP7-4101; B, late juvenile species, MDP8-008; C–E, F–H, I–J, K, L–N, adult specimens, catalog numbers are MDP8-1301, MDP8-007, MDP8-801, MDP8-020 and MDP8-028, respectively. (O–Q) The P₁ element only has one pair of anterior marginal denticles, MDP8-022. (R–Y) Simple ontogenetic series; R–S, juvenile species, MDP8_i052; T–V, W–Y, adult species, catalog no. is MDP8_i050 and MDP8_i046, respectively. Z–BB, variant species, MDP8_i038.   
samples of this taxa collected from MDP8 also allows us to reconstruct the ontogenetic series and to study intraspecific variations, which are presented mainly in Figs. 7–9.

The most common form (Figs. 3RR–3SS, 7A–7C, 8A–8T) resembles the holotype in having a wide platform with one posterior margin deflecting toward the other side near a broadly rounded platform termination. The ontogenetic series of this form (Figs. 8A–8T) reveals that the platform simultaneously posteriorly elongates and laterally expands during the growth. The posterior platform is narrower than the anterior platform during the early growth stage (= in juvenile specimens) and gradually laterally expands until the width of posterior platform is similar to or slightly wider than the anterior platform as the element grows. An apparent identification feature of this form through all the growth stages is that a sudden deflection from the sub-straight platform margin to the pointed platform end occurs near the platform termination, which becomes more distinct as the element grows larger.

The other common form (Figs. 9A–9Y) is characterized by a smooth, rather than sudden, deflection of one platform margin from the anterior to the posterior. Commonly, the platform after the anterior marginal denticles forms a concave outer margin and a convex inner margin which may culminate in a sub-straight edge as the element laterally expands during the late growth stage. The ontogenetic series (Figs. 9A–9L, 9R–9Y) all show a gradual bending of the lateral platform margins after the anterior marginal denticles and the curvature of the carina after the cusp. The outer anterior margin of this form may bear one to three denticles, which increase in height toward the posterior.

Variations observed in rare morphotypes include: (1) a narrow-rounded platform end rather than typical pointed platform end and relatively sub-straight posterior carina and keel (Figs. 8U–8Y); (2) development of small nodes on the deflected posterior margin and platform end to form a denticulated platform termination (Figs. 8Z–8EE); (3) anteriorly increased marginal denticles with one lateral margin bearing four high denticles and the other margin having two denticles (Figs. 9Z–9BB); (4) retention of a weakly asymmetrically bifurcated keel and many fused and continuously aligned lower carinal nodes that extend to the sub-middle platform end (Figs. 7D–7F); and (5) possession of two denticles on each anterior platform margin (Figs. 7G–7O, 7DD).

*Comparison*—Adult *M. zapfei* is easy to differentiate from *M. postera* (*Kozur & Mostler, 1971*) by its larger size, longer platform, and distinct curved carina that extends to or is aligned with the pointed platform end. *M. zapfei* in its early growth stage is very similar to *M. postera* in that the curvature of the carina and the deflected posterior platform margin is not as pronounced as those for its adult elements. However, the juvenile specimens of *M. zapfei* recovered from the Madoupo section either have a carina that extends to the platform end or have an apparently curved posterior platform and posterior carina; and these features contrast with the straight posterior carina that never reaches the posterior platform end of *M. postera*. Because the juvenile *M. zapfei* specimens are easily confused with *M. postera*, caution should be exercised when identifying them. The *M. zapfei* morphotype with a narrowly rounded platform end (Figs. 8U–8Y) resembles *M. slovakensis*, but is differentiated by its broad deflected posterior termination and the

posterior blade which fuses with a low carinal node on the platform and hence has a moderate rather than an abrupt step down to the platform. That morphotype may be an intermediate form between *M. slovakensis* and *M. zapfei*. The variant morphotype with a bifid keel (Figs. 7D–7F) differs from the *M.* aff. *zapfei* in *Yamashita et al. (2018)* by the weakly bifurcated keel end, the upward arched lateral profile as well as more anterior marginal denticles.

*Remarks*—The most common form of the $P_1$ elements almost perfectly matches the description and holotype of *M. zapfei* presented by *Kozur (1972, 1973)*. Other common forms and rare morphotypes also display the common characteristics as described by *Kozur (1972)*. No additional detailed descriptions of *M. zapfei* with figures have been presented in papers since the report of the holotype (*Channell et al., 2003*; *Rigo et al., 2016*; *Karádi et al., 2020*; *Jin et al., 2022*). The only description of *M. zapfei* other than the holotype has no corresponding figures (*Mazza, Rigo & Gullo, 2012*). The abundant specimens collected from level MDP8 allow us to present a detailed illustrated description as well as the field of variation for this species. Intraspecies variations are mainly in the denticulation of anterior platform margins, in the characteristics of the deflection in one posterior platform margin and the posterior carina, and in the features of posterior platform termination or end. *M. zapfei* has been discovered in North America (*Orchard, 1983*), in the Tethyan realm (*Kozur, 1972, 1973*; *Wang & Dong, 1985*; *Channell et al., 2003*; *Hornung, 2005*; *Balini et al., 2010*; *Mazza, Rigo & Gullo, 2012*; *Du et al., 2021*; *Jin et al., 2022*; this study), in Timor (*Martini et al., 2000*), and probably in Japan (*Yamashita et al., 2018*), thereby indicating a global distribution.

*Occurrence*—Sevatian in the Nanshuba Formation of the Madoupo section, Baoshan city, China. Uppermost Alaunian in the Dashuitang Formation of the Dapingdi section (*Wang & Dong, 1985*) and the Hongyan-B section, Baoshan city, China (*Jin et al., 2022*). Sevatian of the Pizzo Mondello section in Sicily, Italy (*Du et al., 2021*). Sevatian in the Rappoltstein section of Rappoltstein Block, southern Germany (*Hornung, 2005*). Sevatian in the Trench section of Silická Brezová, Slovakia (*Channell et al., 2003*). Middle? Alaunian in the Noe Bihati section of West Timor (*Martini et al., 2000*). Middle Alaunian in the Pardonet Formation of the McLay Spur section in British Columbia, Canada (*Orchard, 1983*). Upper Alaunian of Sommeraukogel, Austria (*Kozur, 1972, 1973*).

*Stratigraphic range*—Middle Alaunian to middle Sevatian.

*Mockina* sp. A

Figures 7P–7T
2005 *Epigondolella postera* (*Kozur & Mostler, 1971*); Bazzucchi et al., fig. 11.1.
2016 *Mockina zapfei* (Kozur); Rigo et al., fig. 3.1.

*Materials*—Three $P_1$ elements from MDP8.

*Description*—The $P_1$ elements are characterized by a deflected posterior platform relative to the anterior platform, which is smooth and bears a rounded platform end. The $P_1$

elements are widest on the anterior platform where the marginal denticles grew. The outer platform begins to taper after two marginal denticles, while the inner platform starts to slightly and gradually expand until or near the posterior termination and then tapers to the posterior termination. The anterior platform possesses two high denticles and one highest denticle respectively on the two lateral margins. The blade generally consists of five to six half-fused denticles, with some denticles being very wide. The denticles of the blade is high in the middle and lower on both ends and hence gradually or moderately descends to the low carina toward the posterior. The low carina is commonly comprised of five nodes. The cusp located on the anterior platform is slightly anterior to the position of the posterior-most marginal denticles, and is followed by three carinal nodes. The posterior carina deflects inward and extends to or beyond the platform termination. The pit is located beneath the cusp and within a wide keel. The keel end can be slightly wavy, blunt or narrowly rounded. In lateral view, the basal edge is upwardly stepped beneath the transition between the blade and the low carina.

*Comparison*—*Epigondolella carinata* develops a terminal denticle on the platform end and lateral marginal denticles on two sides of the posterior platform. *M. elongata* has a pointed platform end and a narrower posterior platform. *M. slovakensis* has a blade that is abrupt in the posterior end and consists of highly-fused denticles. *M. medionorica* has similar rounded or narrowly rounded posterior platform end and denticulation of the anterior platform, but can be distinguished by the straight posterior carina which never reaches the posterior end as well as the sub-symmetric posterior platform. *M. postera* has a pointed platform end, an asymmetric posterior platform and a straight carina that doesn't extend to the posterior platform end.

*Remarks*—The $P_1$ elements are similar with *M. zapfei* (Figs. 8U–8Y). But the deflection pattern presented in the three specimens is different from that of *M. zapfei*. The platform margin after two anterior marginal denticles smoothly tapers to the platform end in the $P_1$ elements, contrasting with the corresponding sub-straight margin of *M. zapfei*. The other platform margin after one high anterior marginal denticle always presents a distinct deflection at the posterior fourth of the platform in *M. zapfei*, which is not displayed in the $P_1$ elements. The $P_1$ elements have the whole posterior platform variably deflected toward the inner side relative to the anterior platform, whereas most specimens of *M. zapfei* have only one posterior margin deflected toward the outer side near the platform end.

*Occurrence*—Lower Sevatian (*M. bidentata* Zone in this study) in the Nanshuba Formation of the Madoupo section, China. Sevatian 1 in the Scisti Silicei Formation of the Pignola-Abriola, Italy (*Bazzucchi et al., 2005*; *Rigo et al., 2016*).

*Stratigraphic range*—Sevatian.

Genus *Parvigondolella* *Kozur & Mock, 1972*

*Type species*—*Parvigondolella andrusovi* *Kozur & Mock, 1972* from upper Sevatian of Bohúňovo, Slovakia.

*Parvigondolella andrusovi* Kozur & Mock, 1972
Figures 6TT–6UU
1972 *Parvigondolella andrusovi* n. gen. n. sp.; Kozur & Mock, p. 5, pl. 1, figs. 11, 12.
1974 *Parvigondolella andrusovi* Kozur & Mock, 1972; Kozur & Mock, pl. 1, figs. 11, 12.
1979 *Parvigondolella andrusovi*; Gaździcki et al., pl. 5, figs. 8, 9.
1980 *Parvigondolella andrusovi*; Kovács & Kozur, pl. 15, fig. 3.
1980 *Epigondolella bidentata* Mosher; Krystyn, pl. 14, figs. 5, 6.
1990 *Epigondolella bidentata* Mosher, 1968; Budurov & Sudar, 1990 pl. 5, fig. 12.
2003 *Parvigondolella andrusovi*; Channell et al., pl. A2, figs. 50, 60; pl. A3, figs. 1, 2, 73, 81, 82.
2005 *Parvigondolella andrusovi*; Bertinelli et al., fig. 4/6.
2005 *Parvigondolella andrusovi*; Rigo et al., figs 5.5, 5.6.
2007 *Epigondolella* sp; Pálfy et al., 2007 fig. 6.14.
2009 *Parvigondolella andrusovi*; Rožič et al., fig. 9/g.
2010 *Parvigondolella andrusovi*; Giordano et al., fig. 3.4.
2010 *Parvigondolella andrusovi*; Balini et al., pl. 4, fig. 10.
2012 *Parvigondolella andrusovi*; Mazza et al., p. 126, pl.7, fig. 14.
2012 *Parvigondolella andrusovi*; Gale et al., fig. 4/O.
2018 *Parvigondolella andrusovi*; Yamashita et al., p. 189, fig. 10.4.
2020 *Parvigondolella andrusovi*; Karádi et al., figs. 2D–E, 5T, U.
2021 *Parvigondolella andrusovi*; Du et al., fig. 3. 6.
2021 *Parvigondolella andrusovi*; Zeng et al., figs. 1/e, 4.3.

*Material*—One specimen from POT13.

*Description*—A single blade consists of seven partially fused denticles. The most anterior denticle is small and erect; other denticles incline posteriorly and decrease in height toward the posterior end. The cusp is penultimate and wider than other denticles, and located at the posterior third of the element. The pit is located below the cusp. The groove is anteriorly shallow and narrow; while the keel end is prolonged and narrowly rounded.

*Comparison*—It differs from *Parvigondolella* sp. (described below) in its widest cusp. *M. bidentata* has a reduced and short platform with a pair of marginal denticles or is small in size in early juvenile elements. *P. lata* bear a terminal cusp and fewer denticles. *P. ciarapicae* has a terminal cusp which is the largest denticle of the blade.

*Remarks*—The $P_1$ element has seven denticles, a widest cusp which is situated at the posterior third of the element, and a base with distinct pit, posterior keel and shallow groove. These morphological features match well with the illustration of *P. andrusovi* by Kozur & Mock (1972, pl. 1, fig. 12).

*Occurrence*—Upper Sevatian of Silická Brezová, Slovakia (Kozur & Mock, 1972, 1974; Gaździcki, Kozur & Mock, 1979; Kovács & Kozur, 1980). Sevatian of Sommeraukogel, Austria (Krystyn, 1980). Upper Sevatian of the Kavur Tepe section, Antalya, Turkey and the Scheiblkogel section in Austria (Channell et al., 2003). Sevatian of the Pizzao Mondello

section in Italy (*Balini et al., 2010*; *Mazza, Rigo & Gullo, 2012*). Upper Sevatian in the Calcari con Selce Formation of the Pignola-Abriola section, Mt. S. Enoc section and Mt. Volturino section in the Lagonegro Basin, Italy (*Giordano et al., 2010*; *Karádi et al., 2020*). Sevatian to the Rhaetian in the Calcari con Selce Formation—transitional interval of the Sasso di Castalda section in the Lagonegro Basin, Italy (*Bertinelli et al., 2005*; *Rigo et al., 2005*; *Giordano et al., 2010*). Upper Sevatian to the Rhaetian of the Mt. Kobla section in the Slovenican Basin (*Rožič, Kolar-Jurkovšek & Šmuc, 2009*; *Gale et al., 2012*). Upper Sevatian of Inuyama area, Japan (*Yamashita et al., 2018*). Sevatian in the Nanshuba Formation of the Xiquelin section in Baoshan, China (*Zeng et al., 2021*).

*Stratigraphic range*—Upper Sevatian to Rhaetian.

*Parvigondolella*? *vrielyncki* *Kozur & Mock, 1991*

Figures 6KK–6NN
1980 *Epigondolella bidentata* Mosher; Krystyn, pl. 14, fig. 4.
1991 *Parvigondolella*? *vrielyncki* sp. nov. Kozur and Mock, p. 276–277.
?2003 *Parvigondolella vrielyncki*; Channell et al., pl. A2, fig. 49.
?2003 *Parvigondolella andrusovi* Kozur and Mock; Channell et al., pl. A2, fig. 59, pl. A3, figs. 72, 73, 80.
?2007a *Parvigondolella* sp. B; Orchard et al., p. 363, fig. 7.17.
?2007b *Parvigondolella* sp. C; Orchard et al., pl. 1, fig. 14, 15.
?2018 *Parvigondolella* aff *vrielyncki*; Yamashita et al., p. 189, fig. 10.5.

*Materials*—One $P_1$ element from POT1 and one $P_1$ element from POT13.

*Description*—The $P_1$ elements have a single long blade composed of 10 denticles. The anterior-most one denticle is small and erect, the others slightly inclined and the inclination gets bigger posteriorly. The anterior blade is large and high and decreases in height toward the posterior end. The cusp is the antepenultimate denticle, which is slightly wider than other denticles and situated at the posterior third of the unit. The height of the cusp is lower than its anterior denticles and higher than the posterior two closely positioned denticles. The basal furrow is narrow and shallowly excavated, extends to both ends of the unit, and expands laterally near the position beneath the cusp. In lateral view, the anterior basal edge is straight, while the posterior basal edge slightly bends downward, thereby forming a weakly arched shape.

*Remarks*—The groove in the $P_1$ elements opens and widens along the entire unit to form a narrow and shallow furrow. However, the bases in *M. bidentata*, transitional form between *M. bidentata* and *P. andrusovi* (Figs. 6PP–6QQ, 6VV–6AAA), and *P. andrusovi* are not apparently opened, widened and excavated; thereby enabling the keel, the pit (or basal cavity) and the narrow groove to be still clearly discerned. *Kozur & Mock (1972)* did not illustrate lower view of the holotype of *P. andrusovi* nor did they provide a clear description or diagnosis of that portion. Because most later conodont workers regarded the basal cavity of *P. andrusovi* as not indistinct, we also adopted this feature of identification.

**Table 3 Reported conodont form species in the *Mockina bidentata* Zone.**

| | |
|---|---|
| E. abneptis | *Huckriede (1958)*, *Mosher (1968)*, *Krystyn (1980)*, *Wang & Dong (1985)* |
| E. englandi | *Orchard (1991b)*, *Orchard et al. (2007b)*, *Krystyn et al. (2007)*, *Onoue et al. (2018)*, *Wang et al. (2019)*, *Du et al. (2020)*, *Karádi et al. (2021)* |
| E. aff. englandi | *Wang et al. (2019*, figs. 6.2–6.3), This study |
| E. carinata | *Orchard (1991b)*, *Carter & Orchard (2007)*, *Orchard et al. (2007b)*, *Du et al. (2020)*, this study |
| E. passerii | *Krystyn (1980*, pl. 13, figs. 17 and 18), *Jin et al. (2022)*, this study |
| E. spiculata | *Yamashita et al. (2018)* |
| M. elongata | *Yamashita et al. (2018)*, *Jin et al. (2022*, fig. 4.8), this study |
| M. longidentata | *Kovács & Kozur (1980)* |
| M. mosheri | *Kovács & Kozur (1980)*, *Krystyn et al. (2007)*, *Yamashita et al. (2018)*, *Du et al. (2020)*, *Karádi et al. (2021)*, *Jin et al. (2022)* |
| M. medionorica | *Kovács & Kozur (1980*, pl. 4, fig. 5), this study |
| M. postera | *Kozur & Mostler (1972)*, *Kovács & Kozur (1980)*, *Krystyn (1980)*, *Wang & Dong (1985)*, *Gullo (1996)*, *Channell et al. (2003)*, *Muttoni et al. (2004)*, *Dong & Wang (2006)*, *Rožič, Kolar-Jurkovšek & Šmuc (2009)* |
| M. sakurae | *Zeng et al. (2021)* |
| M. slovakensis | *Gullo (1996)*, *Giordano et al. (2010)*, *Muttoni et al. (2004)*, *Dong & Wang (2006)*, *Rigo et al. (2018)*, *Yamashita et al. (2018)*, *Du et al. (2021)*, *Jin et al. (2022)* |
| M. cf. slovakensis | *Channell et al. (2003)* |
| M. zapfei | *Channell et al. (2003)*, *Hornung (2005)*, *Giordano et al. (2010)*, *Rigo et al. (2018)*, *Du et al. (2021)*, this study |
| M. cf. zapfei | *Bazzucchi et al. (2005)*, *Rigo et al. (2016)* |
| N. steinbergensis | *Mosher (1968)*, *Kovács & Kozur (1980)*, *Krystyn (1980)*, *Orchard (1991b)*, *Channell et al. (2003)*, *Hornung (2005)*, *Orchard et al. (2007b)*, *Rožič, Kolar-Jurkovšek & Šmuc (2009)*, *Mazza, Rigo & Gullo (2012)*, *Onoue et al. (2018)*, *Du et al. (2020)* |
| P.? vrielyncki | *Channell et al. (2003)*, *Rigo et al. (2018)*, *Du et al. (2021)*, this study |
| P. aff. vrielyncki. | *Yamashita et al. (2018)* |
| P. lata | *Du et al. (2021)* |
| $P_1$ elements of Parvigondolella | *Orchard et al. (2007b)*, *Orchard (1991b)*, e.g. |
| Conical $P_1$ elements | *Channell et al. (2003)*, *Du et al. (2021)*, this study |
| Transitional forms from M. bidentata | *Karádi et al. (2020)*, *Du et al. (2021)*, *Zeng et al. (2021)*, this study |

*M. carinata*? in *Du et al. (2020)*, *E. triangularis*? in *Channell et al. (2003)*, *O.? multidentata* in *Kovács & Kozur (1980)*, *M.* aff. *tozeri* and *E. uniformis*? in *Onoue et al. (2018)*, other indeterminated $P_1$ elements.

Alternatively, it is possible the two specimens are a morphotype of *P. andrusovi*. This difficulty in comparing to other taxa indicates the necessity to emphasize the development of the lower side of conodonts in that this view reveals the evolutionary process and thus plays a decisive role in separating the species and genera for Late Triassic conodont elements. Because species of genus *Parvigondolella* have a similar basal field as *M. bidentata*, which contrasts with the wholly covered basal furrow in these two specimens, then it is interpreted that the two specimens do not belong to genus *Parvigondolella*.

On the other hand, even though *Krystyn (1980)* only presented a lateral view for what became the holotype of *P.? vrielyncki* by *Kozur & Mock (1991)*, the diagnosis mentioned a narrow basal furrow and an indistinct basal cavity and *Kozur & Mock (1991)* were also uncertain on its assignment to a genus. The basal features of these $P_1$ elements conform to the description of the holotype of *P.? vrielyncki*. Moreover, the straight edge on the anterior and slightly downwardly bending posterior edge in lateral view of the $P_1$ elements conform to the holotype of *P.? vrielyncki* (*Krystyn, 1980*, pl. 14, fig. 4). Therefore, on the basis of the basal furrow, lateral profile of basal edge and 10 denticles, the two $P_1$ elements are interpreted to be *P.? vrielyncki*. Four specimens presented by *Channell et al. (2003*, fig. A2, 59; figs. A3, 72, 73, 80) and two specimens displayed by *Orchard et al. (2007a*, pl. 1, figs. 14, 15; *2007b*, fig. 7.17) have very similar blades and lateral views as the two $P_1$ elements, and all bear an antepenultimate cusp. However, those authors did not illustrate the basal cavities of those specimens, therefore a full comparison is not possible. Another *P.* aff. *vrielyncki* identified in *Yamashita et al. (2018)* has a crest-shaped blade and arched lateral view; but again, no lower view was presented, therefore it is not certain whether their *P.* aff. *vrielyncki* is identical with these two $P_1$ elements.

*Occurrence*—Sevatian of the Potou section in Baoshan city, China. Alaunian 2/IV (= upper *Halorites macer* Zone, which had been regarded as lower Sevatian by *Kozur & Mock, 1991*) of western Timor, Indonesia (*Krystyn, 1980*, pl. 14, fig. 4).

*Stratigraphic range*—Upper Alaunian to Sevatian.

## DISCUSSION

Conodont fauna in the *Mockina bidentata* Zone of the Sevatian are probably very diverse. There are at least 19 forms of $P_1$ elements (see Fig. 2, Table 2), which belong to six form genera within the *M. bidentata* Zone of the Potou section and the Madoupo section. This reveals a high taxonomic diversity and might suggest a peak in diversification of conodonts during the early Sevatian. In the first part of this article, the review of Sevatian conodont biostratigraphy gives a detailed account of the range of reported conodont species within the *M. bidentata* Zone. Our current compilation, which is only preliminary, of the reported conodont form species within the *M. bidentata* Zone (Table 3) suggests that there may be more than 30 different forms of $P_1$ elements occurring in the *M. bidentata* Zone. Many known species that occurred during middle Norian actually range upward into the Sevatian, such as *E. spiculata*, *M. elongata*, *M. medionorica*, *M. postera* and *E. abneptis* with no-bifid keel end. It can be inferred that other middle Norian conodonts may have survived into the Sevatian. The occurrence of ornated $P_1$ elements (*E. abneptis*, *E. triangularis*?, *E. uniformis*? and *M.* aff. *tozeri*) in the *M. bidentata* Zone also implies a more diverse conodont fauna was present in the Sevatian world than previously thought. More new species or new forms of $P_1$ elements have been discovered in the *M. bidentata* Zone in recent years, which indicates that the peak of diversity that *Plasencia, Aliaga & Sha (2013)* assigned as end-Norian actually occurred during the *M. bidentata* Zone of the early Sevatian.

However, this peak in diversity during the *M. bidentata Zone* was followed by a biotic crisis during the middle of the Sevatian. The termination of conodont species ranges within the Potou section imply an apparent rapid decline in conodont diversity above sample layer POT9, and most of the higher horizons yielded only single species. This implies that there was a conodont crisis (= major decline in diversity) in the uppermost *M. bidentata* Zone or near the boundary between *M. bidentata* Zone and *P. andrusovi* Zone. The zonal boundary interval is characterized by transitional forms between *M. bidentata* and *P. andrusovi*, by the disappearance of most segminiplanate gondolellid conodonts, and by the appearance of species of *Parvigondolella (Parvigondolella* sp.*)*.

Indeed, also within the western Tethys regions, only very few conodont species occur within the *P. andrusovi* Zone. The associated assemblages are commonly *P. lata-N. steinbergensis* (*Gaździcki, Kozur & Mock, 1979*), single *M. bidentata* (*Rigo et al., 2005*), *M. slovakensis-M. bidentata-N. steinbergensis* in the Pignola-Abriola section and *P. lata-M. bidentata-P. vrielyncki* in Mt. S. Enoc and Mt. Volturino sections (*Giordano et al., 2010*), *M. slovakensis-M. zapfei* (*Channell et al., 2003*) with *M. bidentata-N. steinbergensis* (*Balini et al., 2010*; *Mazza, Rigo & Gullo, 2012*), and *M. bidentata-M.* ex gr. *postera-N. steinbergensis* (*Rožič, Kolar-Jurkovšek & Šmuc, 2009*; *Gale et al., 2012*). These global records suggest that only four segminiplanate conodont species of *M. slovakensis*, *M. bidentata*, *M. zapfei* (with the exception of the unclear *M.* ex gr. *postera*) and *N. steinbergensis* survived from the *M. bidentata* Zone into the *P. andrusovi* Zone and that segminate conodonts (*P. lata, P. ciarapicae etc.*) commenced their radiation in the *P. andrusovi* Zone (*Karádi et al., 2020*). This is in contrast with the diverse conodont species discovered in the *M. bidentata* Zone of the Baoshan area (*Wang et al., 2019*; *Du et al., 2020*; *Zeng et al., 2021*; *Jin et al., 2022*) and of other early Sevatian strata around the world (Table 3). Therefore, it is obvious that conodonts suffered a significant crisis in the middle of the Sevatian.

*Rigo et al. (2018)* interpreted that an important bioevent corresponding to the presence of conodont genus *Parvigondolella* occurred during a warm phase (W3) of late Norian (*Trotter et al., 2015*). The progression of transitional form between *M. bidentata* and *P. andrusovi* (Figs. 6PP–6QQ, 6VV–6AAA) and *Parvigondolella* sp. recovered from the Potou section below the first occurrence of *P. andrusovi* allows a refined conodont biostratigraphy with an interval of transitional forms between the *M. bidentata* Zone and the *P. andrusovi* Zone that corresponds to this bioevent.

It is recommended that future studies should focus on this transitional interval spanning the uppermost *M. bidentata* Zone to onset of the *P. andrusovi* Zone to understand the underlying causes of this first crisis that heralded the end-Triassic conodont extinction interval and the relationships of conodont evolution to environment changes.

## CONCLUSIONS

Conodont faunas were recovered from upper Norian (Upper Triassic) strata of the Dashuitang and Nanshuba formations in two sections near Baoshan City in western Yunnan Province. These assemblages provide important insights on the biostratigraphy

and diversity history of Norian conodonts in China as well as throughout the Tethyan realm. Conodont *Mockina* (*M.*) *bidentata* Zone and *Parvigondolella* (*P.*) *andrusovi* Zone are identified in this area, thereby greatly improving the resolution of the Sevatian biochronostratigraphy in the Baoshan area and enabling a more precise correlation with other sections around the world. In total, 19 different forms of $P_1$ elements, which belong to six different form genera, are found within the *M. bidentata* Zone of lower Sevatian. When these are combined with reported form species in the same conodont zone, a peak in conodont diversity is revealed in the *M. bidentata* Zone. This peak was followed by a distinct decline in conodont diversity and a pronounced morphologic change and turnover in the uppermost *M. bidentata* Zone, which may greatly help to shed light on the study of the first crisis of the protracted suite of end-Triassic mass extinctions.

## ACKNOWLEDGEMENTS

We thank Y. F. Gong and Q. Cao for field assistance in sampling. SEM pictures were taken at the State Key Laboratory of Biogeology and Environmental Geology in China University of Geosciences and at the Wuhan Center of Geological Survey. C.B. Yan is appreciated in helping with the SEM photographs. Special thanks to M. Rigo, X. L. Lai and to the reviewers M. Leu, M. L. Golding and V. Karádi for their suggestions on conodont taxonomy, and to editor Kenneth De Baets for his useful comments during review process.

### Funding

This work was supported by the National Natural Science Foundation of China (Grant numbers 41830320, 41972033). The funders had no role in study design, data collection and analysis, decision to publish, or preparation of the manuscript.

### Grant Disclosures

The following grant information was disclosed by the authors:
National Natural Science Foundation of China: 41830320, 41972033.

### Competing Interests

The authors declare that they have no competing interests.

### Author Contributions

- Weiping Zeng conceived and designed the experiments, performed the experiments, analyzed the data, prepared figures and/or tables, authored or reviewed drafts of the article, and approved the final draft.
- Haishui Jiang conceived and designed the experiments, analyzed the data, prepared figures and/or tables, authored or reviewed drafts of the article, and approved the final draft.
- Yan Chen analyzed the data, authored or reviewed drafts of the article, and approved the final draft.

- James Ogg analyzed the data, authored or reviewed drafts of the article, and approved the final draft.
- Muhui Zhang performed the experiments, analyzed the data, prepared figures and/or tables, and approved the final draft.
- Hanxinshuo Dong performed the experiments, analyzed the data, prepared figures and/ or tables, and approved the final draft.

## Data Availability

The raw data are available in the Supplemental File.

## Supplemental Information

Supplemental information for this article can be found online at http://dx.doi.org/10.7717/peerj.14517#supplemental-information.

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
