# Peer review of "Upper Norian conodonts from the Baoshan block, western Yunnan, southwestern China, and implications for conodont turnover"

_PeerJ, doi:10.7717/peerj.14517_

## Round 0.1 · original submission · Major Revisions

You provide an important documentation and study of conodonts from the Norian of South China crucial for a better understanding of the geographic and stratigraphic distribution of various species as well as regional differences in morphology. I would like to see this work published but there are some crucial aspects which need revision:

Biostratigraphy: the conodont associations of the sections is consistent with a lower Sevatian 1 (lower part of the upper Norian) rather than close to the top of the Norian based on conodont associations as typical Misikella elements are missing (see reviewer 3). In this context, it is also necessary to rephrase and reduce large parts of the text which is focused on the Norian/Rhaetian boundary or NRB as well as the Carnian/Norian boundary or CNB (see reviewer 3). Your conodont findings are largely irrelevant to these issues and therefore should be only briefly mentioned if at all.

Taxonomy: you describe nine new species based on limited material. The definition for various most species is poorly justified as multiple species fall within the intraspecific variation of others or represent juvenile specimens of known species. Only one of those is likely justified (compare reviewer 3). The erection of less species would not diminish the importance of your study. My recommendation would be to assign the ones which fall clearly within the variation of previous described species to those species rather than erecting new species where appropriate under open nomenclature. In case of uncertainties, I would be commendable to rather describe them under open nomenclature than erecting new species that are difficult to justify or are junior synonyms of previously described species (compare reviewer 1)

Geological Setting: you are repeating many aspects referring to the geological setting (see reviewers 1 and 2) in the introduction (lines 178-218) as well as in the Material and Methods sections (Lines 218-260). It would be more appropriate to have a single dedicated geological setting section or paragraph.

Phylogenetic reconstruction and evolutionary trends: the phylogenetic reconstruction needs to be re-evaluated excluding invalid taxa which would make it more like previous work which needs to be more appropriately referenced (reviewer 3). This will also affect the interpreted evolutionary trends as M. monodentata is a juvenile form of M. bidentata (compare reviewer 1)

Discussion: the whole discussion part including the evolutionary trend with M. monodentata should be revised and re-written because M. monodentata is simply a juvenile form of M. bidentata and therefore many hypotheses on the evolutionary trends are invalid (compare reviewers 1 and 3). The trends have been previously documented (compare reviewer 3) and should be more appropriately referenced. A brief discussion would still be relevant as it would confirm the evolution between Mockina and Parvigondolella to be global and further corroborates the importance of genus Parvigondolella not only for phylogenetic purpose but also for biostratigraphy (compare reviewer 3)



Cited references: I agree with reviewer 1 that Gupta et al. 1980 would best be removed or replaced as reference. As pointed out by reviewer 3, the introduction and discussing of the (bio)stratigraphic assignment (also compare reviewer 2) and phylogeny misses multiple new and relevant studies.

Formatting/Language/Typographical issues: the language of manuscript is mostly clear and unambiguous but there is room for further improvement (see reviewer 1 and 2).

I greatly apologise for the delay in my decision but obtaining sufficient reviews and comparing them with each other took longer than I expected. I look forward to receiving the revised manuscript. Please address these points as well as all other points raised by the reviewers including those in the annotated pdfs.

·

Basic reporting

The English is mostly clear and unambiguous throughout the manuscript but could be improved.

Line 111: Gupta et al. 1980 should be removed or replaced as reference. Gupta was known of distortion of results. See: Webster, G. D., Rexroad, C. B., & Talent, J. A. (1993). An evaluation of the VJ Gupta conodont papers. Journal of Paleontology, 67(3), 486-493.

Line 121: “Oddly” is wrong wording. Main reason for the lack of studies for the past 20 years is based on the limited access to this region for many researchers. This is not “odd”. Better words could be “Unfortunately”,“However” or “Nevertheless”

Line 152: "It is obvious the lower and middle Norian..." change in the grammar is needed

Lines 174-175: If these Norian and Rhaetian conodonts were already found in this regions in the past, then is it not a "probability" but rather a "certainty" to find Norian and Rhaetian conodonts in this region? Maybe better to write "...indicating a great probability of re-finding upper Norian to Rhaetian strata"

Lines 178-218 and 221-249. There is no geological setting part. However, the authors write part of the geological setting within the introduction (lines 178-218) and another part within the Material and Methods (lines 221-249). There are some repetitions within the second part from the first part. Better would be to merge both parts together to one Geological setting part.

Experimental design

Conodont faunas from the upper Norian in southwest China are rarely documented and neglected over the last 20 years. Therefore this research is relevant and meaningful not only for the conodont community but for all Triassic workers in terms of stratigraphic intercalibration and global biotic turnover events at the end of the Triassic.

The research was performed with the modern standard methods and described with sufficient detail.

Validity of the findings

The authors claim to describe nine new conodont species. However, this claim should be rejected or re-written for most of the new species for the following reasons:

326 Epigondolella inordinata n. sp.

Only 3 specimens were found. For a formal description of a new species, at least 5-10 specimens should be discoverd. Would be better to keep this taxon in open nomenclature. Furthermore, the holotype seems to be a broken element. In addition, all illustrated specimens are rather small and could be juvenile forms of E. humboldtensis. (No specimens of E. humboldtensis were identified from the samples for comparison).

Line 371 Epigondolella longienglandi n. sp.

Only 5 specimens were found which is the bare minimum for a profound determination of a new species. In this case, a morphological extremly similar species (E. englandi) is already established and the main difference seems to be of regional nature, it would make more sense to determine it for the moment as E. aff. englandi until more specimens are found. (No specimens of E. englandi were identified from the samples for comparison).


Line 422 Epigondolella madoupoensis n. sp.

Here again, only 5 specimens were discovered. The main morphological difference to E. carinata is the size. This is insufficient. Why can it not simply be that these specimens are large forms of E carinata? (no E. carinata were identified in the samples for comparison)


Line 462 Epigondolella potouensis n. sp.

Enough specimens to determine a new species including a synonymy list for comparison. E serrulata is the closest similar form but also couldnt be found in any of the recovered samples.


Line 646 Mockina deflecta n. sp.


Only 3 specimens are found. Should be kept in open nomenclature until more specimens are found. Could be intraspecific variation of Mockina zapfei.

Line 697 Mockina monodentata n. sp.

Probably only an intraspecific variation of M. bidentata. Only 6 specimens were recovered. The specimens are all relatively small and therefore there is a high chance that these specimens ar simply juvenile forms of M. bidentata where the second denticle is not yet developed.

Line 739 Mockina tridentata n. sp.

Occurs in the same zone as morphological very similar M. bidentata. Only 3 elements are found. All specimens are not very well preserved. With only 3 specimens it is difficult to say if this is intraspecific variation, pathological forms or simply unidentifiable because of the poor preservation.


Line 1075-1084 It has to be questioned how likely it is to find so many “new” taxa from a not very abundant amount of samples and specimens which coincidentally are morphological all extremely similar to already existing Tethyan taxa. I agree, that the regional variation should be taken into account but not to this degree of creation new species out of such few data. Maybe “aff.” if the authors are certain of a new species but lack the numbers (>10 specimens). Maybe creating asubspecies could be an option but I am afraid that most specimens are too poorly preserved. Otherwise if already existing species exist with a high morphological similarity and occur in the same stratigraphic zone, the possibility of intraspecific variation should be strongly considered in this case.

Discussion: the whole discussion part including the evolutionary trend with M. monodentata should be revised and re-written because M. monodentata is simply a juvenile form of M. bidentata and therefore many hypothesis on the evolutionary trends are invalid.

1098-1100 I disagree with this interpretation. It is not a downsizing and the loss of a denticle. M. monodentata is simply a juvenile form of M. bidentata where the marginal denticle is not yet fully developed.

Additional comments

I am highly concerned with the taxonomical part of this manuscript. Furthermore should some parts of the introduction and the material and methods merged into a seperate chapter "Geological settings"

·

Basic reporting

There are minor spelling and grammar mistakes throughout the paper, which I have not attempted to correct here. References are adequate, and the paper is well-illustrated.

Experimental design

The research question is clearly defined, and it is clearly demonstrated how the data answers this question. The methodology is appropriate and rigorously applied.

Validity of the findings

The conclusions are valid and based clearly on the data presented; this paper is novel and will be of use to a variety of researchers in the field.

Additional comments

I have reviewed a previous version of this paper for another journal, and found it worthy of publication with minor revisions then, and I do not see any reason to change that recommendation. The current version does need some revision. In addition to improvements to spelling and grammar (perhaps at the copy editing stage), I have also noticed the following:

Line 42 - 50: The CNB discussion needs to be update with reference to the paper of Hounslow et al. 2021 in Albertiana, which outlines the latest decisions on assigning proxy and type section for this boundary.

Line 178 - 218: This section would be more suited to a Geological Setting section than the Introduction.

Line 218 - 260: This is just a repetition of lines 178 - 218.

Line 1001: As the following section is distinct from the Systematic Paleontology, a new sub-heading is required.

Line 1078, 1079: typos - inordinata and elongata.

Line 1170: typo - potuensis.

Acknowledgments - I suggest include Rigo and Karadi here, as they provided input on previous version s of the manuscript that have been incorporated into the present version.

Reviewer 3 ·

Basic reporting

Dear Editor PhD Dr. De Baets
I have received the request to review the manuscript (#68099) entitled "New conodont faunas from the upper Norian (Late Triassic) of western Yunnan, southwestern China." by Zeng and co-authors.


I have already reviewed this manuscript twice for a different journal, and the Authors prefer to re-submit the text, figures, plates, and tables without any changes. I thus reported the same revisions, with some adjustments.

I strongly recommend the Authors to consider the revisions of the Referees.

The age of the stratigraphic sections is Sevatian 1, based on the conodont association. Since there are no conodonts of Sevatian 2, it is not correct to state that the section is close to the Norian/Rhaetian boundary, cause the described conodont association missed Misikella elements. So they are still Sevatian 1 (e.g. Kozur and Mock, 1991), base of the upper Norian. The Authors should carefully read at least the papers by Wotzlaw et al. (2014), Kent et al. (2017), Maron et al. (2019) for the age calibrations of the Upper Triassic stages.

The authors are not aware of some important Norian conodont papers. Actually they are aware, cause they cited 2 important papers Orchard et al., 2007 and Carter and Orchard 2007, but they disregarded the distributions of the illustrated conodonts (see replies below). In fact, conodonts M. spiculata, M. carinata, M. elongata and M. englandi are not limited to middle Norian, but they are documented and thus occurred also in the Sevatian, and M. carinata and M. englandi even in the RHAETIAN. For instance:
- M. spiculata and M. elongata occur within the M. bidentata Zone (upper Norian, Sevatian) in Japan as illustrated in Yamashita et al., 2015. Please note that Yamashita’s paper was reviewed also by M. Rigo, an anonymous referee and M. Orchard (experts of Upper Triassic conodonts), who first described the M. spiculata and M. elongate, thus accepting the Sevatian age of these 2 species;
- M. carinata (and also M. englandi) was also found both in the bidentata (Sevatian 1), in the Misikella hernsteini-Parvigondolella andrusovi zone (Sevatian 2) but also in the RHAETIAN as documented for instance in Orchard et al. 2007; Orchard and Carter 2007. Please note also here that M. Orchard first described M. carinata and M. englandi too, thus extending the age of the species to Rhaetian.
Some of species have been also found in upper Norian beds (i.g. Sevatian) for instance in western Tethys (e.g. Krystyn et al., 2007) and in China (Du et al. 2020), that means globally.

The line 1075-1084 is an easy way to skip the real issue of this manuscript that is the description of not real new species. The new species described are:
- transition forms between 2 species (e.g. M. monodentata),
- intraspecific variation of already described species (e.g. E. longienglandi)
- juvenile forms (e.g. the holotype of M. boashanensis)

The authors thus intentionally omitted important stratigraphic distributions of well documented and cosmopolitan Norian conodonts. The intention of the authors is to describe several new species, forcing the phylogenetic relationships/interpretations without any scientific evidence, ignoring the literature. Most of the new species, which are not real, are also juvenile elements as clearly illustrated by Mazza & Martínez-Pérez (2015) and the authors do not consider the intraspecific variations of the species.


Below the detailed revision submitted on 14th September 2020:
1. The first part of the introduction is not related to the topic of the manuscript. The studied section is lower Sevatian in age but the introduction is mostly about the Carnian/Norian boundary (CNB) and on the Norian/Rhaetian boundary (NRB), which are like 20 Myr after or 10 Myr before the Sevatian. Furthermore, the CNB is not described as in detail as the NRB. The NRB is not well described as well, missing a lot of important citations (e.g. there are 2 proposed markers for the NRB but only 1 is discussed; new cycle-magneto calibrations; new phylogenetic papers about NRB conodonts) and the issue about M. posthernsteini has already been solved and well discussed in Bertinelli et al., 2016 and Karadi et al., 2020, and previously in Giordano et al. 2010, and Rigo et al., 2016.
Throughout the manuscript in fact, the Norian/Rhaetian boundary appears to be the main topic, but the collected data are not related to the NRB. It seems that the Authors try to move the discussion towards a more debated topic.
I recommend the Authors to delete the NRB and CNB part from introduction and keep only the second part. Furthermore, the Authors need to add the correct citations of previous articles, especially in the introduction which is a state of the art. I added some of the most important but other should be added

2. I carefully checked the description of the new species. The Authors actually found only 1 new species, that is Mockina potouensis. All the other “new” species has to be included in the morphological variability or within the population of already-described species. The Authors seem being not familiar with the paper by Mazza and Martìnez-Pérez (2015 – Bollettino della Società Paleontologica Italiana), in which they show the ontogenetic processes in the Upper Triassic conodonts through growth series reconstructions and X-ray microtomography. In their paper, Mazza and Martìnez-Pérez proved that different morphologies are strictly related to different ontogenetic processes and that certain morphologies can be recognized already from the most juvenile growth stages. The “new” species described in this manuscript correspond exactly to those variations due to the ontogenetic processes and to the population variability of each species. For instance, the length of the free blade is too much related to the ontogenetic stage (longer in juvenile stage and shorter in adult or late adult stage) to be considered a good feature to recognize a new species (like longienglandi). In fact, the length of the blade has been abandoned since the early ‘90s. Moreover, a better features might be the number of the denticles on the free blade, but again the number falls within the intraspecific variation, such as Epig. englandi described and illustrated by Orchard 1991 that show the same number of the denticle in the free blade of E. longienglandi.
The other aspect that has to be considered when describing a new taxon is related to the forms belonged to the morphocline between the ancestor and the descendant species. In fact, the morphocline represents all the transitional forms between mother and daughter species, which is commonly used for paleozoic conodonts and described also for Triassic conodonts by Giordano et al., 2010, and successively by Mazza et al., 2012 and Kàradi et al., 2020, the latter by using a cladistic-statistical approach (the same used for other vertebrate fossils like dinosaurs, or brids, or mammals). The new approach on modern conodont research adopted by Triassic conodont workers is not to split species unless is necessary but instead to recognize their population, avoiding to create more confusion on the taxonomy and palaeontology. Many researchers oversimplified conodont taxonomy and deeper studies should be done before establishing new species.

3. Paleontological part:

- Epigondolella inordinata
The description of this new species is based on only 3 specimens (including one juvenile, one broken), all of them illustrated in plate 3, 21-28:
Specimens 21-23 is almost juvenile. The Authors in fact described that the prolonged keel end may be not pronounced in juvenile specimens, so they recognized that 1 specimen is juvenile.
Specimens 27-28 is broken and the lower side is not illustrated.
Specimens 26-24 is the holotype. Actually the holotype is a Mockina spiculata, typical are the outwardly-projected denticles on the posterior margin, the typical pronounced convexity of the outer posterior margin, the keel shape and the position of the pit.

- Epigondolella longienglandi
This species is not a new species, but a real Mockina englandi.
The Authors split E. longienglandi form E. englandi considering the length of the free blade (nearly the half of the element). The holotype is a juvenile form of E. englandi (e.g. the basal groove) and the blade has the same number of denticles of the holotype described by Orchard (1991). Basically, E. longienglandi is a intraspecific variation of the E. englandi described originally by Orchard. In fact, Orchard (1991) illustrated a paratype of E. englandi that looks like longienglandi (table 5, fig 20). It is very interesting that the Authors did not put this paratype in synonym list of E. longiengladi, maybe they didn’t notice the similarity or maybe they consider this paratype as a real E. englandi even if it corresponds to the description of longienglandi. In this case, E. longienglandi would correspond to E. englandi.
Furthermore, the Authors did not consider the intraspecific variation of the blade of E. englandi illustrated by Orchard, which shows a number of denticles of the free blade between 6 to 8, like the number of the denticles of the illustrated specimens of longienglandi. At present, there are no elements to describe this new species.

- Epigondolella madoupoensis
In the remarks the Authors stated that this new species resembles E. carinata described by Orchard (1991) in the marginal denticulation and platform shape, and that only differs in having a larger size and longer blade and free blade, which is the main diagnostic characters, along with 2+1 denticles on the anterior margins. The blade should be between 1/3 to 1/2 of the unit length. Basically the holotype described by Orchard in 1991. In fact, even if Orchard didn’t describe the blade length, it is clearly visible the length on the lateral view, which is actually 1/3 of the length of the unit. Also the 2+1 denticles on the anterior margins are a diagnostic character of E. carinata.
It is not clear why the Authors included specimens 3.1-8 which show more than 2+1 denticles on the anterior margins and but 2 to 4 like in E. tozeri by Orchard 1991.
Conodont 5.25-28 is a juvenile species, that might be E. tozeri, E. carinata or E. bidentata or, more probably, E. elongate.
This is another case of misclassification. This species is not a new one.

- Mockina boashanensis
All the species illustrated are Mockina elongata. The original description by Orchard, 1991 perfectly described all the specimens illustrated by the Authors (5.1-24), including the proposed holotype that is actually a sub-aldut.

- Mockina deflecta
This description falls into the intraspecific variability of Mockina slovakensis. For instance, the Authors put specimens 6.7-9 as M. slovakensis, but it can be either M. deflecta.

- Mockina monodentata
Thesse specimens are the transitional forms between M. bidentata and P. andrusovi, as illustrated and discussed by serval authors and a big literature is available. In the remarks, the Authors confirmed that is a transitional form between the mother and daughter species. Transitional forms are not real species. Moreover, the designated holotype is clearly a juvenile form.

- Mockina tridentata
these forms belong to the M. potuoensis species, and they are sub-adult forms. Only 3 specimens have been found but only 1 illustrated. Better included in the M. potuoensis population


Zieglericonus: the 2 elements (4.21-23) classified as Zieglericonus are instead fragment of ramiforms

Norigondolella: 3.39-40 is a blade of a platform conodonts. 3.41-42 is probably a Norigondolella, better put with question mark


4. Discussion.
This part basically is the description of the trends described recently by Karadi et al. 2020 (published online in 2019), who also applied a cladistic-statistical approach. The difference is that the Authors described M. monodentata as a new species, the holotype of which is a juvenile form, which is also a transitional form of the morphocline between Mockina and Parvigondolella. Even fig 10 is similar to fig 3 by Karadi et al., 2020.
This discussion doesn’t bring any new information about the phylogenetic relationship or trend between the Mockina and Parvigondolella. Instead it confirmed that the evolution between genera Mockina and Parvigondolella is global and it testifies the importance of genus Parvigondolella not only for phylogenetic purpose but also for biostratigraphy.

Experimental design

no comment

Validity of the findings

no comment

Additional comments

no comment

---

## Round 0.2 · Minor Revisions

Thank you for addressing our previous suggestions including describing some of the specimens under open nomenclature which has made the manuscript more consistent and easier to follow. I would love to this work with valuable descriptions and illustrations published, but there are still some issues in the revised version which need to be addressed before publication. The main points being:

Open nomenclature: Personally, I appreciate the description and illustration of specimens in open nomenclature. I agree with reviewer 3 that Mockina n sp. A and Parvigondolella n. sp. A are likely not new species but due to preservation or lack of additional material the author might express their opinion that specimens are hard to assign to existing species – this happens and falls to the authors describing the material to convey the rationale behind this decision. As opposed to the reviewer 3, I feel such species could still be reported in open nomenclature (without adding n. sp.) if their potential similarity and differences from known specimens/species is properly highlighted and discussed. Subsequent author might have a different opinion and could still revise or assign those specimens – if there are not mentioned/figured at all such a valuable discourse cannot happen. Reviewer 1 also pointed out that specimens described as M. aff. zapfei do not resemble zapfei at all and might be better described as a separate species in open nomenclature.

Use of Epigondolella: reviewer 1 pointed out some discrepancies in the description of Epigondolella and Mockina. Just clearly state which author(s) and rationale you are following in these assignments and be consistent in its use.

Assignments to other species: various assignment to other species (e.g., englandi, carinata, bidentata, elongata, zapfei, postera) need to be further scrutinized and discussed (compare reviewer 1).

Conodont biostratigraphy and biogeography: Please make sure that in other sections previous records and species assignments are equally scrutinized critically as they are in the synonymy lists. The previously reported occurrence of multidentata in China are likely not multidentata and restricted to North America (compare Reviewer 1). Please make sure an updated taxonomy is used here too.

Species ranges: the age range of medionorica and its comparison with multidentate needs to be better justified (compare reviewer 1). Please also make sure the full range of carinata is reported (compare reviewer 1).

Typographical and language issues: I share the opinion of reviewer 1 that several issues were introduced in the revision. Please make sure those are rectified before resubmission (compare reviewer 1).

Please make sure these and all other points raised by the reviewers (also those in attached pdfs) are addressed.

I look forward to receiving the revised manuscript.

·

Basic reporting

The authors have made substantial revisions to the text, as recommended by the reviewers. However, this has introduced numerous errors in English spelling, grammar and usage, which will have to be rectified prior to publication. The use of language in the paper is not suitable for publication as it stands. I recommend that one of the authors who is proficient in English re-read the paper prior to re-submission.

Otherwise, the literature survey, figures, and data presentation are all suitable, and the conclusions are supported by the data.

Experimental design

The paper follows the appropriate methodology for a biostratigraphic study.

Validity of the findings

The authors have significantly revised their taxonomy based on the recommendations of other reviewers. I understand why they have done this, although I previously supported their attempts to identify additional species. I have several comments regarding the new taxonomy:

1. Use of Epigondolella.
The authors use this genus for the species carinata and englandi. Both of these species clearly belong to the genus Mockina; the lobate platform outlines, smooth posterior platform margins, and strong carina that reaches the posterior end of the platform are all consistent with this genus. Epigondolella, in contrast, possesses stronger posterior platform ornamentation and a carina that does not reach the posterior of the element. The type species of Mockina, M. postera, and other typical species such as M. matthewi, illustrate the similarity between these species and M. carinata and M. englandi. I recommend the authors reconsider the inclusion of these species in Epigondolella.

2. Definition of englandi.
In several places it is stated that M. englandi differs from M. carinata in having "symmetrically arranged marginal denticles." This is not the definition of englandi. Instead, the species is based on those forms which possess only two, high, denticles on the anterior of the platform (one higher than the other). There are often accessory denticles, much smaller in size, to the posterior.

3. Range of carinata.
It is erroneously stated that carinata was for a long time only reported in North America, and the only other localities listed are in China. Carinata has also been identified in the Alps (Donofrio et al 2003) and Japan (Ishida and Hirsch 2001).

4. Use of old species names.
I found at least on example of a species name in the paper that the authors introduced in a previous version, but have now cut: e.g. longienglandi (line 449). There may be others, please check.

5. M. aff. englandi
The authors state that the only signifcant change between this species and M. englandi, is a longer blade; however, it also possesses more numerous posterior margin denticles, and a bifurcated keel that is highly unlike that of englandi. This requires more discussion.

6. M. bidentata.
Several specimens in this species are described as "probably a new species with affinity to M. bidentata" In which case they should be aff. bidentata!

7. M. elongata.
These are far to short and broad to be elongata (only 5G really resembles this species). How do these specimens differ from what you are calling carinata?

8. Age range of medionorica.
The author suggest that because Kovacs and Kozur illustrate a specimen of medionorica as multidentata, the age range of these species must be the same. I do not understand the logic of this statement. How do the authors know the scope of the species that Kovacs and Kozur were considering to belong to multidentata?

9. M. zapfei.
It would be best to show the specimens in size order if the authors wish to call it a growth series. It is not clear with specimens of different sizes spread across the plate. Also, zapfei cannot be considered "a good marker species for global correlation" if it ranges from the middle Alaunian to middle Sevatian, a very long range.

10. M. aff. zapfei.
These specimens do not resemble zapfei at all, as they lack the sinuous platform, carina and basal keel. It would be better to keep these as a separate species in open nomenclature.

11. zapfei vs. postera.
This is very tricky; the type of zapfei is a juvenile, whereas the type of postera is an adult. It is therefore unclear to me that these really are two separate species. The carina of specimens referred to postera does reach the end of the platform, in disagreement with the authors (e.g. see Orchard 1991). If there is no way to separate the species, and they are simply different growth stages, then postera has priority.

12. P. vrielyncki.
The figure caption labels more specimens than indicated in the figure list.

Additional comments

I also have some suggestions for other parts of the paper:

1. Development of Sevatian conodont biostratigraphy.
In contrast to the synonymy lists, wherein previous records have been scrutinized critically, and species assignments changed, in this section occurrences are taken at face value. For example, the occurrence of multidentata in China; these are not multidentata, which appears to be restricted to North America. I would like to see this section edited to include updated taxonomy of the regions discussed.

2. Discussion.
There is a contradiction when the authors say "this supports the opinion of Martinez-Perez that there were no major declines in conodont diversity in the late Norian", followed by "a biotic crisis during the middle of the Sevatian". Do the authors think there was a decline in the late Norian, or not?

Reviewer 3 ·

Basic reporting

See comments on the attached pdf file

Experimental design

See comments on the attached pdf file

Validity of the findings

See comments on the attached pdf file

Additional comments

See comments on the attached pdf file

Annotated reviews are not available for download in order to protect the identity of reviewers who chose to remain anonymous.

---

## Round 0.3 · Minor Revisions

Thank you for your patience and addressing the final mostly taxonomic requests by the reviewers. These changes have made the manuscript even more consistent and easier to follow. There is no need for further peer review, but I discovered some remaining language and formatting issues which I would like to see addressed before publication. I look forward to receiving the revised manuscript and finally able to see this comprehensive work published. All points can be found in the annotated pdf to make the process more efficient.

---

## Round 0.4 · accepted · Accept

Thank you for addressing all final suggestions which makes the manuscript even easier to follow. I agree with the all changes you addressed. I look forward to seeing this work published.